# Robust Bayesian Regression via Hard Thresholding

**Zheyi Fan**[1,2], **Zhaohui Li**[3†], **Qingpei Hu**[1,2†]

[1]Academy of Mathematics and Systems Science, Chinese Academy of Sciences, China
[2]School of Mathematical Sciences, University of Chinese Academy of Sciences, China
[3]H. Milton Stewart School of Industrial and Systems Engineering, Georgia Institute of Technology, USA.
[1,2]{fanzheyi,qingpeihu}@amss.ac.cn, [3]zhaohui.li@gatech.edu

## Abstract

By combining robust regression and prior information, we develop an effective robust regression method that can resist adaptive adversarial attacks. Due to the widespread existence of noise and data corruption, it is necessary to recover the true regression parameters when a certain proportion of the response variables have been corrupted. Methods to overcome this problem often involve robust least-squares regression. However, few methods achieve good performance when dealing with severe adaptive adversarial attacks. Based on the combination of prior information and robust regression via hard thresholding from [1], this paper proposes an algorithm that improves the breakdown point when facing adaptive adversarial attacks. Furthermore, to improve the robustness and reduce the estimation error caused by the inclusion of a prior, the idea of Bayesian reweighting is used to construct a more robust algorithm. We prove the theoretical convergence of proposed algorithms under mild conditions. Extensive experiments show that, under different dataset attacks, our algorithms achieve state-of-the-art results compared with other benchmark algorithms, demonstrating the robustness of the proposed approach.

## 1 Introduction

Least-squares methods are widely used because of their simplicity and ease of operation. However, due to the inevitable existence of outliers, least-squares methods, such as linear regression, may cause significant bias in practical applications. Therefore, to meet the challenge of learning reliable regression coefficients in the presence of significant corruption in the response vector, this paper focuses on robust least-squares regression (RLSR). RLSR has excellent application value in many fields, such as signal processing [8][12][26], economics [23], industry [21], biology [13], remote sensing [11] and intelligent transportation [25].

Given a data matrix $X = [\mathbf{x}_1, ..., \mathbf{x}_n] \in \mathbb{R}^{d \times n}$, the corresponding response vector $\mathbf{y} \in \mathbb{R}^n$, and a certain number $k$ representing the number of corruptions in the data, the RLSR problem can be described as:

$$(\hat{\mathbf{w}}, \hat{S}) = \arg \min_{\substack{\mathbf{w} \in \mathbb{R}^p, S \subset [n] \\ |S| = n-k}} \sum_{i \in S} (y_i - \mathbf{x}_i^T \mathbf{w})^2 \tag{1}$$

That is, we aim to recover the correct point set $S$ and the regression coefficient $\mathbf{w}^*$ simultaneously to achieve the minimum regression error. However, this problem is NP hard, so it is difficult to optimize directly [19].

To solve the above problem, a commonly used data generation model is $\mathbf{y} = X^T \mathbf{w}^* + \mathbf{b}^* + \boldsymbol{\epsilon}$, where $\mathbf{w}^*$ is the true regression coefficient we wish to recover and $\boldsymbol{\epsilon}$ is a dense white noise vector

---

[†]Corresponding authors.

36th Conference on Neural Information Processing Systems (NeurIPS 2022).

subject to a specific distribution, that is, $\|\boldsymbol{\epsilon}\|_0 \sim n$. The vector $\mathbf{b}^*$ is $k$-sparse, which means that there are only $k$ non-zero values, representing $k$ unbounded noise terms in the response vector. After years of development, there are many ways to find a reasonable solution to the problem in Eq. (1). However, these methods typically only achieve good performance under specific conditions. The main challenge is the low breakdown point of conventional methods. The breakdown point $k$ is a measure of robustness, which means the number of corruptions that the RLSR algorithm can tolerate. We can express $k$ as the proportion of all data points: $k = \alpha \cdot n$. Many RLSR algorithms cannot guarantee theoretical convergence as the value of $k$ increases. For example, McWilliams et al. [14] used weighted subsampling for linear regression, but only had a breakdown point of $\alpha = O(1/\sqrt{d})$. Prasad et al. [18] proposed a robust gradient estimator that can be applied to linear regression, but their method only tolerates corruption up to $\alpha = O(1/\log d)$. Other methods may have a higher breakdown point, but tend to assume a specific pattern of data corruption. One representative adversary model for introducing data corruption is the oblivious adversarial attack (OAA), in which the opponent generates $k$ sparse vectors while completely ignoring $X$, $\mathbf{w}^*$, and $\boldsymbol{\epsilon}$. The work of Bhatia et al. [1] and Suggala et al. [20] reported excellent results against OAAs by using a novel hard thresholding method; indeed, [20] suggested that $\alpha$ may even get close to 1 as $n \to \infty$. The recent online fast robust regression algorithm [16] also has consistent convergence with a mild condition by using Stochastic Gradient Decent (SGD) algorithm. However, these methods cannot resist adaptive adversarial attack (AAA), in which opponents can view $X$, $\mathbf{w}^*$, and $\boldsymbol{\epsilon}$ before determining $\mathbf{b}^*$. Handling AAA is a challenging task, and so many methods can only guarantee a very low breakdown point, especially when the data distribution is not normal [4][9][15]. Bhatia et al. [2] proposed the thresholding operator-based robust regression method and their breakdown point reach $1/65$ for noiseless model i.e., $\boldsymbol{\epsilon} \equiv 0$. However, their method can give a consistent estimation only in noiseless case. Karmalka et al. [10] had a good result in sparse robust linear regression by applying the $L_1$ regression and the breakdown point of their method reaches 0.239, but their estimation is consistent only when white noise $\boldsymbol{\epsilon}$ is sparse. Diakonikolas et al. [6] considered the situation that $X$ and $\mathbf{y}$ may have outliers simultaneously, and proposed a filter algorithm in which the error bound is $O(\alpha \log(1/\alpha)\sigma)$. However, their method requires accurate data covariance of the true data distribution or numerous unlabeled correct data to estimate the data covariance, which are often unavailable in practice.

The limitations of the above-mentioned methods can be attributed to a lack of prior knowledge from the real data, making it difficult to distinguish the set of correct points in the case of AAAs. Gülçehre et al. [7] showed that prior information effectively improves the accuracy of machine learning. In many application scenarios in industry, economics, and biology, prior knowledge such as previous experimental data or engineering data are available. The goal of this paper is to propose a new robust regression that can integrate available prior information, even if the prior information is not very accurate.

The typical approach for integrating prior information is the Bayesian method. This provides a way of formalizing the process of learning from data to update beliefs in accordance with recent notions of knowledge synthesis [5]. However, generic Bayesian method is also sensitive to outliers. Thus robust Bayesian method should be considered to produce more reliable estimates in the presence of data corruption. Polson et al. [17] used the local variance to assign each point a local parameter that makes the estimation result robust. Furthermore, Wang et al. [22] proposed a local parameterization method and used empirical Bayesian estimation to determine the global parameters. Bhatia et al. [3] proposed a Bayesian descent method using an unadjusted Langevin algorithm (ULA), which guarantees convergence in a finite number of steps. Wang et al. [24] employed Bayesian reweighting to assign different weights to samples, thus reducing the impact of outliers.

In this paper, we combine the Bayesian method with a hard thresholding method [1] and propose two algorithms, which we call TRIP and BRHT. Through assigning a simple normal prior on the coefficients, TRIP can significantly increase the breakdown point when resisting AAAs. To further improve the accuracy of estimation, we propose BRHT algorithm through applying Bayesian reweighting method [24] to coefficient estimation. Experiments show that BRHT is even more resistant to AAAs and gives lower estimation errors, demonstrating that our method achieves significantly improved robustness.

**Our Contributions**: The main contribution of this paper is proposing new methods that combining the prior and robust regression to increase the breakdown point when encountering AAAs. We derive the theoretical guarantees given by our proposed algorithms. Compared with the consistent robust

regression (CRR) algorithm proposed in [1], we prove that our algorithms guarantee convergence under a weaker condition, which also shows that our methods improves the breakdown point. We also establish an extended experiment to test the effectiveness of the algorithms. Compared with other basic algorithms, the experimental results show that our methods significantly outperform alternative methods under AAAs. Moreover, BRHT algorithm is also competitive against OAAs.

**Paper Organization**: We state the problem formulation and present some notation and tools in Section 2. In Section 3, we describe the details of our proposed TRIP and BRHT algorithms. The theoretical properties of these two algorithms are discussed in Section 4. Section 5 presents extensive experimental results that demonstrate the excellent performance of our proposed algorithms. Section 6 concludes this paper.

## 2 Problem Formulation

In this study, we mainly focus on the problem of RLSR under AAAs. We are given a covariant matrix $X = [\mathbf{x}_1, ..., \mathbf{x}_n] \in \mathbb{R}^{d \times n}$, where $\mathbf{x}_i \in \mathbb{R}^d$. The true coefficient of the regression model is denoted by $\mathbf{w}^*$. The response vector $\mathbf{y} \in \mathbb{R}^n$ is generated by:

$$\mathbf{y} = X^T \mathbf{w}^* + \mathbf{b}^* + \boldsymbol{\epsilon} \tag{2}$$

The perturbations to the response vector consist of two parts: the adversarial corruption vector introduced by $\mathbf{b}^*$, which is a $k$-sparse vector, and the dense white noise $\epsilon_i \sim \mathcal{N}(0, \sigma^2)$. Our goal is to recover the true regression coefficient $\mathbf{w}^*$ while simultaneously determining the corruption set $S$. To illustrate our problem, we first pay attention to the standard robust regression problem in Eq. (1). From the viewpoint of probability, the problem can be transformed into a log-likelihood version:

$$(\hat{\mathbf{w}}, \hat{S}) = \arg \max_{\substack{\mathbf{w} \in \mathbb{R}^p, S \subset [n] \\ |S| = n-k}} \sum_{i \in S} \log \ell(\mathbf{w} \mid y_i, \mathbf{x}_i, \sigma^2) \tag{3}$$

We will try to convert the problem in Eq. (3) into a Bayesian version. From the Bayesian viewpoint, we consider $p_\mathbf{w}(\mathbf{w})$ as the prior of the coefficients in the model. In addition, we add the localization parameter $\mathbf{r}$ and its prior $p_\mathbf{r}(\mathbf{r})$ to reflect the change introduced by each additional sample. For any subset $S \subseteq [n]$, the distribution of all parameters and data $X_S, \mathbf{y}_S$ is:

$$p(\mathbf{y}_S, \mathbf{w}, \mathbf{r}_S \mid X_S) = p_\mathbf{w}(\mathbf{w}) p_\mathbf{r}(\mathbf{r}_S) \prod_{i \in S} \ell(y_i \mid r_i, \mathbf{w}, \mathbf{x}_i, \sigma^2) \tag{4}$$

The posterior distributions $p(\mathbf{w}, \mathbf{r}_S | X_S, \mathbf{y}_S)$ and $p(\mathbf{y}_S, \mathbf{w}, \mathbf{r}_S | X_S)$ differ by a regularization constant. We ignore this regularization constant in the posterior distribution and only consider the main terms of the parameters. We then formulate the Bayesian RLSR problem of searching for the subset and coefficients by maximizing the log-posterior:

$$(\hat{\mathbf{w}}, \hat{S}) = \arg \max_{\substack{\mathbf{w} \in \mathbb{R}^p, \mathbf{r} \in \mathbb{R}^n_+ \\ S \subset [n], |S| = n-k}} \log p_\mathbf{w}(\mathbf{w}) + \sum_{i \in S} [\log \ell(y_i \mid r_i, \mathbf{w}, \mathbf{x}_i, \sigma^2) + \log p_\mathbf{r}(r_i)] \tag{5}$$

Note that we do not add any prior on $\sigma^2$ and only treat this as an adjustable parameter. This is because, in the initial stage of the algorithm described in Section 3, the estimated $\sigma^2$ will be large due to the existence of outliers, and this will make the estimation of $\mathbf{w}$ excessively biased to the prior distribution. This bias will be harmful, especially when the prior is not sufficiently accurate. This phenomenon can be observed in Section 3.1. To prove the positive effect of the prior on the RLSR problem, we require the properties of *Subset Strong Convexity (SSC)* and *Subset Strong Smoothness (SSS)*. Given a set $S \subset [n]$, $X_S := [\mathbf{x}_{i \in S}] \in \mathbb{R}^{d \times |S|}$ signifies the matrix with columns in the set $S$. The smallest and largest eigenvalues of a square symmetric matrix $X$ are denoted by $\lambda_{min}(X)$ and $\lambda_{max}(X)$.

**Definition 1** (SSC Property). *A matrix $X \in \mathbb{R}^{d \times n}$ is said to satisfy the SSC property at level $m$ with constant $\lambda_m$ if the following holds:*

$$\lambda_m \leq \min_{|S|=m} \lambda_{min}(X_S X_S^T) \tag{6}$$

**Definition 2** (SSS Property). *A matrix $X \in \mathbb{R}^{d \times n}$ is said to satisfy the SSS property at level $m$ with constant $\Lambda_m$ if the following holds:*

$$\max_{|S|=m} \lambda_{max}(X_S X_S^T) \leq \Lambda_m \tag{7}$$

---

**Algorithm 1 TRIP**: hard **T**hresholding approach to **R**obust regression with s**I**mple **P**rior

---

**Input:** Covariates $X = [\mathbf{x}_1, ..., \mathbf{x}_n]$, responses $\mathbf{y} = [y_1, ..., y_n]^T$, prior knowledge $\mathbf{w}_0$,
   penalty matrix $M$, corruption index $k$, tolerance $\epsilon$
**Output:** solution $\hat{\mathbf{w}}$
1: $\mathbf{b}^0 \leftarrow \mathbf{0}, t \leftarrow 0,$
   $P_{MX} \leftarrow X^T(XX^T + M)^{-1}X, P_{MM} \leftarrow X^T(XX^T + M)^{-1}M$
2: **while** $\|\mathbf{b}^t - \mathbf{b}^{t-1}\|_2 > \epsilon$ **do**
3:    $\mathbf{b}^{t+1} \leftarrow HT_k(P_{MX}\mathbf{b}^t + (I - P_{MX})\mathbf{y} - P_{MM}\mathbf{w}_0)$
4:    $t \leftarrow t + 1;$
5: **end while**
6: **return** $\hat{\mathbf{w}} \leftarrow (XX^T)^{-1}X(\mathbf{y} - \mathbf{b}^t)$

---

These two properties are proposed in [2], and are intended to standardize the generation of the data matrix so that it will not be too abnormal. They are used to prove the theorems in Section 4.

## 3   Methodology

We first ignore the localization parameter $\mathbf{r}$ in Eq. (5) and propose a simple method called TRIP in Section 3.1. TRIP demonstrates the effect of a prior on the hard thresholding method. To improve the robustness and the accuracy of the estimation, the BRHT algorithm is proposed in Section 3.2.

### 3.1   TRIP: Hard Thresholding Approach to Robust Regression with Simple Prior

We propose a robust regression algorithm called TRIP (Algorithm 1), a hard Thresholding approach to Robust regression with sImple Prior. In this subsection, only the prior $p_{\mathbf{w}}(\mathbf{w})$ is considered and the localization parameter $\mathbf{r}$ is not added to the model. We assume the variance $\sigma^2$ of $\epsilon_i$ can be set by ourselves and that the prior $p_{\mathbf{w}}(\mathbf{w})$ obeys a normal distribution $\mathcal{N}(\mathbf{w}_0, \Sigma_0)$, where $\mathbf{w}_0$ and $\Sigma_0$ are determined in advance. Through the above simple parameter settings, the problem in Eq. (5) is transformed into the problem in Eq. (1) with an additional regularization term:

$$(\hat{\mathbf{w}}, \hat{S}) = \arg \min_{\substack{\mathbf{w} \in \mathbb{R}^p, S \subset [n] \\ |S| = n-k}} \sum_{i \in S}(y_i - x_i^T\mathbf{w})^2 + (\mathbf{w} - \mathbf{w}_0)^T M(\mathbf{w} - \mathbf{w}_0) \tag{8}$$

where $M = (\Sigma_0/\sigma^2)^{-1}$. To solve this problem, we are motivated by the hard thresholding method proposed by Bhatia [1], which concentrated on recovering the errors instead of selecting the 'cleanest' set. The problem in Eq. (8) can be formulated as $\min_{\mathbf{w} \in \mathbb{R}^p, \|\mathbf{b}\|_0 \le k^*} \frac{1}{2}\|X^T\mathbf{w} - (\mathbf{y} - \mathbf{b})\|_2^2 + \frac{1}{2}(\mathbf{w} - \mathbf{w}_0)^T M(\mathbf{w} - \mathbf{w}_0)$. Thus, if we have an estimation $\hat{\mathbf{b}}$ of the corruption vector $\mathbf{b}^*$, the estimation of $\mathbf{w}^*$ can be easily obtained by $\hat{\mathbf{w}} = (XX^T + M)^{-1}[X(\mathbf{y} - \hat{\mathbf{b}}) + M\mathbf{w}_0]$. By substituting this estimation into the optimization problem, we obtain a new formulation of the problem:

$$\min_{\|\mathbf{b}\|_0 \le k^*} f(\mathbf{b}) = \frac{1}{2}\|(P_{MX} - I)(\mathbf{y} - \mathbf{b}) + P_{MM}\mathbf{w}_0\|_2^2 \tag{9}$$

where $P_{MX} = X^T(XX^T + M)^{-1}X$, $P_{MM} = X^T(XX^T + M)^{-1}M$. The hard thresholding step in the TRIP algorithm can be viewed as $\mathbf{b}^{t+1} = HT_k(\mathbf{b}^t - \nabla f(\mathbf{b}^t))$, where $k$ is the selected corruption coefficient. The hard thresholding operator $HT_k$ is defined as follows.

**Definition 3** (Hard Thresholding). *For any vector $\mathbf{r} \in \mathbb{R}^n$, let $\delta_{\mathbf{r}}^{-1}(i)$ represent the position of the $i^{th}$ element in $\mathbf{r}$, which are arranged in descending order of magnitude. Then, for any $k < n$, the hard thresholding operator is defined as $\hat{\mathbf{r}} = HT_k(\mathbf{r})$, where $\hat{\mathbf{r}}_i = \mathbf{r}_i$ if $\delta_{\mathbf{r}}^{-1}(i) \le k$ and $0$ otherwise.*

The difference between the proposed TRIP algorithm and the original CRR [1] is the form of iteration step. The iteration step in both TRIP and CRR can be expressed uniformly as $HT_k(\mathbf{y} - X^T\mathbf{w}^t)$, but $\mathbf{w}^t = (XX^T + M)^{-1}[X(\mathbf{y} - \mathbf{b}^t) + M\mathbf{w}_0]$ in TRIP and $\mathbf{w}^t = (XX^T)^{-1}X(\mathbf{y} - \mathbf{b}^t)$ in CRR. The $\mathbf{w}^t$ in CRR is just a simple least square estimation, while the prior added in TRIP can be regarded to adding a quadratic regularization in each iteration. This quadratic regularization can avoid the candidate of iteration that is too far from the prior mean, which is also helpful to ensure the

---

**Algorithm 2 BRHT**: robust **B**ayesian **R**eweighting regression via **H**ard **T**hresholding

---

**Input:** Covariates $X = [\mathbf{x}_1, ..., \mathbf{x}_n]$, responses $\mathbf{y} = [y_1, ..., y_n]^T$, prior distribution $p_{\mathbf{r}}(\mathbf{r}), p_{\mathbf{w}}(\mathbf{w})$
    , corruption index $k$, tolerance $\epsilon$
**Output:** solution $\hat{\mathbf{w}}$
 1: $\mathbf{b}^0 \leftarrow \mathbf{0}, t \leftarrow 0,$
 2: **while** $\|\mathbf{b}^t - \mathbf{b}^{t-1}\|_2 > \epsilon$ **do**
 3:    $\mathbf{w}^t \leftarrow VBEM(X, \mathbf{y} - \mathbf{b}^t, p_{\mathbf{r}}(\mathbf{r}), p_{\mathbf{w}}(\mathbf{w}))$
 4:    $\mathbf{b}^{t+1} \leftarrow HT_k(\mathbf{y} - X^T\mathbf{w}^t)$
 5:    $t \leftarrow t + 1;$
 6: **end while**
 7: **return** $\hat{\mathbf{w}} \leftarrow (XX^T)^{-1}X(\mathbf{y} - \mathbf{b}^t)$

---

**Algorithm 3 VBEM**: **V**ariational **B**ayes **E**xpectation **M**aximization

---

**Input:** Covariates $X = [\mathbf{x}_1, ..., \mathbf{x}_n]$, responses $\mathbf{y} = [y_1, ..., y_n]^T$, prior distribution $p_{\mathbf{r}}(\mathbf{r}), p_{\mathbf{w}}(\mathbf{w})$
**Output:** solution $\hat{\mathbf{w}}$
 1: **repeat**
 2:    update $q(\mathbf{r})$
 3:    update $q(\mathbf{w})$
 4: **until** convergence
 5: **return** $\hat{\mathbf{w}} \leftarrow \text{MAP}(q(\mathbf{w}))$

---

numerical stability of solution. Thus, as long as the prior is not mis-specified too much, TRIP will be more likely to identify the uncorrupted points, and the final result of TRIP will be more robust than CRR.

Therefore, the prior plays an important role in TRIP. However, the weight of a prior in the solution depends entirely on the matrix $M = (\Sigma_0/\sigma^2)^{-1}$. Therefore, if we use an estimation of $\hat{\sigma}^2$ to replace $\sigma^2$, or give $\sigma^2$ a prior to calculate its posterior distribution, the overestimation of $\sigma^2$ will cause a severe increase in $M$ in the initial iteration steps. This will mislead the iteration and cause some deviation in the final results. To overcome this difficulty we can directly treat $M$ as an adjustable parameter to control by specifying the form such as $M = sI$, where $s$ is a positive number, and the suitable parameter can be chosen through 5-fold or 10-fold cross validation.

### 3.2 BRHT: Robust Bayesian Reweighting Regression via Hard Thresholding

In this subsection, we describe how the Bayesian reweighting method is combined with hard thresholding to give a more robust algorithm, BRHT (Algorithm 2), a robust Bayesian Reweighting regression via Hard Thresholding. We first introduce the reweighted probabilistic model (RPM) proposed in [24] for traditional linear regression. For the covariates $X$ and the response $\mathbf{y}$, the RPM model can be formulated as follows:

$$p(\mathbf{y}, \mathbf{w}, \mathbf{r}|X) = \frac{1}{Z}p_{\mathbf{w}}(\mathbf{w})p_{\mathbf{r}}(\mathbf{r})\prod_{i=1}^{n}\ell(y_i \mid \mathbf{w}, \mathbf{x}_i, \sigma^2)^{r_i} \tag{10}$$

where $\mathbf{r}$ is the local weight assigned to each sample, $Z$ is the normalizing constant, $\ell(y_i \mid \mathbf{w}, \mathbf{x}_i, \sigma^2)$ represents the likelihood of the normal distribution $\mathcal{N}(\mathbf{x}_i^T\mathbf{w}, \sigma^2)$, and $p_{\mathbf{w}}(\mathbf{w})$, $p_{\mathbf{r}}(\mathbf{r})$ are the priors of $\mathbf{w}$ and $\mathbf{r}$, respectively. By ignoring the normalizing constant, the problem in Eq. (5) can be transformed into the following form under this RPM setting:

$$(\hat{\mathbf{w}}, \hat{S}) = \arg\max_{\substack{\mathbf{w}\in\mathbb{R}^p, \mathbf{r}\in\mathbb{R}_+^n \\ S\subset[n], |S|=n-k}} \log p_{\mathbf{w}}(\mathbf{w}) + \sum_{i\in S}[r_i \log \ell(y_i|\mathbf{w}, \mathbf{x}_i, \sigma^2) + \log p_{\mathbf{r}}(r_i)] \tag{11}$$

The specific form of the prior $p_{\mathbf{r}}(r_i)$ can be set to any nonnegative random variable distribution, including (but not limited to) the Gamma distribution, Beta distribution, or log-normal distribution. Here, we still use the normal distribution $\mathcal{N}(\mathbf{w}_0, \Sigma_0)$ as the form of $p_{\mathbf{w}}(\mathbf{w})$.

To solve the optimization problem in Eq. (11), we use the two-step BRHT algorithm. The key iteration step in BRHT is $\mathbf{b}^{t+1} \leftarrow HT_k(\mathbf{y} - X^T\mathbf{w}^t)$, where $\mathbf{w}^t$ is calculated by maximizing the

log-posterior of the RPM model:

$$(\mathbf{w}^t, \mathbf{r}^t) = \arg\max_{\mathbf{w}\in\mathbb{R}^d, \mathbf{r}\in\mathbb{R}^n_+} \log p_{\mathbf{w}}(\mathbf{w}) + \log p_{\mathbf{r}}(\mathbf{r}) + \sum_{i=1}^{n} r_i \log \ell(y_i - b_i^t \mid \mathbf{w}, \mathbf{x}_i, \sigma^2) \qquad (12)$$

However, the direct inference of Eq. (12) is hard because of the nonconvexity of this problem. In general, the parameters in this RPM model can be divided into two parts: the global variable $\mathbf{w}$ and the local latent variable $\mathbf{r}$. To solve the inference problem of RPM, a feasible method is to use variational Bayesian expectation maximization (VBEM). We set $q(\mathbf{w}, \mathbf{r}) = q(\mathbf{w})q(\mathbf{r})$ to approximate the true posterior after several iterations of VBEM (Algorithm 3), and replace the estimation of $\mathbf{w}$ by the maximum a posteriori (MAP) estimation from the final approximate posterior. Full details of the VBEM method are given in Appendix A. It is reasonable to ask why we are using Bayesian reweighting. Note that the iteration step in the TRIP algorithm is $\mathbf{b}^{t+1} \leftarrow HT_k(P_{MX}\mathbf{b}^t + (I - P_{MX})\mathbf{y} - P_{MM}\mathbf{w}_0) = HT_k(\mathbf{y} - X^T\mathbf{w}^t)$, where $\mathbf{w}^t = (XX^T + M)^{-1}[X(\mathbf{y} - \mathbf{b}^t) + M\mathbf{w}_0]$. Although we can show that TRIP already guarantees theoretical convergence, the estimation of $\mathbf{w}^*$ in every iteration still uses least-squares with a penalty term, which is easily affected by the corrupted points. This disadvantage forces us to assign a higher weight to the prior to resist severe data corruption in the case of AAAs. However, a higher weight on the prior means a larger estimation bias. When applying the Bayesian reweighting method, the estimation in each step is more robust than the least-squares result, and thus the weight on the prior can be reduced to guide the iteration. Therefore, the estimation bias is relatively small and the results are more robust. This is reflected in the experimental results presented in Section 5.

We should also explain why we only use a prior for a few parameters. It is important to ensure that the prior weights are neither too high nor too low. As mentioned earlier, if we treat $\sigma^2$ as the parameter to be estimated, this places too much weight on the prior. We also ensure that $p_{\mathbf{w}}(\mathbf{w})$ does not create more uncertainty, such as setting $p_{\mathbf{w}}(\mathbf{w})$ to $p(\mathbf{w}, \alpha) = \mathcal{N}(\mathbf{w}_0, \alpha^{-1}\Sigma_0)Gam(\alpha|a_\alpha, b_\alpha)$. Data corruption means that the subset of the training data may vary greatly from the prior information. Thus, when calculating the posterior, the variance of $\mathbf{w}$ controlled by $\alpha$ will be very large to fit the data, and so the prior information $\mathbf{w}_0$ will have lower weights in the inference step. The above problems also mislead the estimation and the selection of subsets, so we do not consider the uncertainty of these quantities and simply treat them as model parameters to be set in advance. An adjustment method for all the parameters in BRHT is described in Appendix D.

## 4 Theoretical Convergence Analysis

In this section, we establish the convergence theory for the TRIP algorithm, and clearly explain how the prior effectively enhances the convergence of the RLSR model. Theorems 1 and 2 summarize the results. We also show the theoretical guarantee of the BRHT algorithm in Theorems 3–5, which further demonstrate the special properties achieved by using Bayesian reweighting. Before presenting the convergence result, we first introduce some notation. Let $\lambda^t := (XX^T + M)^{-1}X(\mathbf{b}^t - \mathbf{b}^*)$, $\mathbf{g} := (I - P_{MX})\epsilon$, and $\mathbf{f} := P_{MM}(\mathbf{w}^* - \mathbf{w}_0)$. Let $S_t := [n]\backslash supp(\mathbf{b}^t)$ be the chosen subset that is considered to be uncorrupted, and $I_t := supp(\mathbf{b}^t) \cup supp(\mathbf{b}^*)$.

**Theorem 1.** *Let $X = [\mathbf{x}_1, \ldots, \mathbf{x}_n] \in \mathbb{R}^{d\times n}$ be the given data matrix and $\mathbf{y} = X^T\mathbf{w}^* + \mathbf{b}^* + \boldsymbol{\epsilon}$ be the corrupted output with sparse corruption of $\|\mathbf{b}^*\|_0 \leq k \cdot n$. For a specific positive semi-definite matrix $M$, $X$ satisfies the SSC and SSS properties such that $2\frac{\Lambda_{k+k^*}}{\lambda_{min}(XX^T+M)} < 1$. Then, if $k > k^*$, it is guaranteed with a probability of at least $1 - \delta$ that, for any $\varepsilon, \delta > 0$, $\|\mathbf{b}^{T_0} - \mathbf{b}^*\|_2 \leq \varepsilon + O(e_0) + O(\frac{\sqrt{\Lambda_{k+k^*}}\lambda_{max}(M)}{\lambda_{min}(XX^T+M)})\|\mathbf{w}^* - \mathbf{w}_0\|_2$ after $T_0 = O(\log(\frac{\|\mathbf{b}^*\|_2}{\varepsilon}))$ iterations of TRIP, where $e_0 = O(\sigma\sqrt{(k+k^*)\log\frac{n}{\delta(k+k^*)}})$ under the normal design.*

**Theorem 2.** *Under the conditions of Theorem 1, and assuming that $\mathbf{x}_i \in \mathbb{R}^d$ are generated from the standard normal distribution, if $k > k^*$, it is guaranteed with a probability of at least $1 - \delta$ that, for any $\varepsilon, \delta > 0$, the current estimation coefficient $\mathbf{w}_{T_0}$ satisfies $\|\mathbf{w}_{T_0} - \mathbf{w}^*\|_2 \leq O(\frac{1}{\sqrt{n}})(\varepsilon + e_0) + O(\frac{\sqrt{k+k^*}\lambda_{max}(M)}{n^{3/2}})\|\mathbf{w}^* - \mathbf{w}_0\|_2$ after $T_0 = O(\log(\frac{\|\mathbf{b}^*\|_2}{\varepsilon}))$ steps.*

For positive semi-definite matrices $XX^T$ and $M$, $\lambda_{min}(XX^T + M) \geq \lambda_{min}(XX^T) + \lambda_{min}(M)$. Thus, the condition $2\frac{\Lambda_{k+k^*}}{\lambda_{min}(XX^T+M)} < 1$ in Theorem 1 is weaker than the condition $2\frac{\Lambda_{k+k^*}}{\lambda_{min}(XX^T)} <$

1 of Lemma 5 of Bhatia [1], which shows that a prior can effectively improve the convergence of the algorithm. Assigning a higher weight to a prior means that $M$ has larger eigenvalues, so that the convergence condition will be more easily satisfied. As a result, the TRIP algorithm can tolerate a higher proportion of outliers than the CRR method of Bhatia [1] and achieves a higher breakdown point. In fact, under the condition $\lim_{n\to\infty} \frac{\lambda_{min}(M)}{n} = \xi$, we can give an approximate expression of the breakdown point for TRIP when $\xi$ is not too large: $k^* \leq k \leq (0.3023 - \sqrt{0.0887 - 0.0040\xi})n$. Details can be found in Appendix C.2. However, the improved convergence comes at the cost of an unavoidable reduction in precision. This can be seen from Theorem 2. If the data corruption is such that $k^*$ is $O(n)$ and the maximum eigenvalue of $M$ is also $O(n)$, then the bias of $\hat{w}$ cannot be decreased by adding more samples, which shows that there is a trade-off between convergence and accuracy. A reliable prior improves both accuracy and convergence because it has a higher weight. However, an inaccurate prior can also be helpful as long as it is quite different from the distribution of outliers, and the convergence can be improved through a prior with a low weight.

To prove the properties of our BRHT algorithm, we define the following two intermediate variables to simplify the description:

$$U(\mathbf{w}, \mathbf{r}, S) = \log p_{\mathbf{w}}(\mathbf{w}) + \sum_{i \in S}[\log p_{\mathbf{r}}(r_i) + r_i \log \ell(y_i \mid \mathbf{w}, \mathbf{x}_i, \sigma^2)] \tag{13}$$

$$M(\mathbf{w}, \mathbf{r}, \mathbf{b}) = \log p_{\mathbf{w}}(\mathbf{w}) + \sum_{i}[\log p_{\mathbf{r}}(r_i) + r_i \log \ell(y_i - b_i \mid \mathbf{w}, \mathbf{x}_i, \sigma^2)] \tag{14}$$

**Theorem 5.** *Suppose that the prior of $r_i$ is independently and identically distributed (iid). We consider the $t^{th}$ iteration step of the BRHT algorithm, where $\mathbf{w}_t, \mathbf{r}_t = \arg\max_{\mathbf{w}\in\mathbb{R}^d, \mathbf{r}\in\mathbb{R}_+^n} M(\mathbf{w}, \mathbf{r}, \mathbf{b}_t)$ and $\mathbf{b}_t = HT_k(\mathbf{y} - X^T\mathbf{w}_{t-1})$ is obtained from the hard thresholding step. Then, we have that $U(\mathbf{w}_t, \mathbf{r}_t, S_{t+1}) \geq U(\mathbf{w}_{t-1}, \mathbf{r}_{t-1}, S_t)$.*

**Theorem 6.** *Consider a data matrix $X$ and a specific positive semi-definite matrix $M$ satisfying the SSC and SSS properties such that $2\frac{\Lambda_{k+k^*}}{\lambda_{min}(XX^T+M)} < 1$. Then, there exist $\alpha > 0$ and $0 < \gamma \leq 1 + \epsilon$, where $\epsilon$ is a small number, such that if $k > k^*$ and $\Sigma$ in the prior $p_{\mathbf{w}}(\mathbf{w})$ is $\alpha\sigma^2 M^{-1}$, it is guaranteed with a probability of at least $1 - \delta$ that, for any $\varepsilon, \delta > 0$, $\|\mathbf{b}^{T_0} - \mathbf{b}^*\|_2 \leq \varepsilon + O(e_0) + O(\frac{\sqrt{\Lambda_{k+k^*}\lambda_{max}(M)}}{\lambda_{min}(XX^T+M)})\gamma\|\mathbf{w}^* - \mathbf{w}_0\|_2$ after $T_0 = O(\log(\frac{\gamma\|\mathbf{b}^*\|_2}{\varepsilon}))$ iterations of BRHT, where $e_0 = O(\sigma\sqrt{(k+k^*)\log\frac{n}{\delta(k+k^*)}})$ under the normal design.*

**Theorem 7.** *Under the conditions of Theorem 4 and with $\mathbf{x}_i \in \mathbb{R}^d$ generated from the standard normal distribution, there exist $\alpha > 0$ and $0 < \gamma \leq 1 + \epsilon$, where $\epsilon$ is a small number, such that if $k > k^*$ and $\Sigma$ in the prior $p_{\mathbf{w}}(\mathbf{w})$ is $\alpha\sigma^2 M^{-1}$, it can be guaranteed with a probability of at least $1 - \delta$ that, for any $\varepsilon, \delta > 0$, the current estimation coefficient $\mathbf{w}_{T_0}$ satisfies $\|\mathbf{w}_{T_0} - \mathbf{w}^*\|_2 \leq O(\frac{1}{\sqrt{n}})(\varepsilon + e_0) + O(\frac{\sqrt{k+k^*}\lambda_{max}(M)}{n^{3/2}})\gamma\|\mathbf{w}^* - \mathbf{w}_0\|_2$ after $T_0 = O(\log(\frac{\gamma\|\mathbf{b}^*\|_2}{\varepsilon}))$ steps.*

Theorem 5 shows that our BRHT algorithm is reasonable because it optimizes the problem in Eq. (11) in each step. Theorems 6 and 7 guarantee the convergence of the parameter, which means that if we introduce a prior $p_{\mathbf{w}}(\mathbf{w}) = \mathcal{N}(\mathbf{w}_0, \Sigma_0)$ to the TRIP algorithm, it is convergent. There then exists some $\alpha > 0$ such that the prior $p_{\mathbf{w}}(\mathbf{w}) = \mathcal{N}(\mathbf{w}_0, \alpha\Sigma_0)$ in the BRHT algorithm guarantees convergent parameters and the bias of the estimation of $\mathbf{w}^*$ will be $\gamma$ times that of TRIP. Note that $\alpha$ is usually relatively large in practice, which causes a lower prior weight in BRHT. Thus, when the convergence can be guaranteed, the bias of the estimator $\hat{w}$ can be significantly reduced because BRHT assigns a lower weight to the prior. Even if the data are seriously corrupted, BRHT can ensure good results without significant error. All proofs of these theorems are given in Appendix C.

## 5 Experiments

In this section, we first consider how to effectively 'corrupt' the dataset using two different attacks: OAA and AAA. We then report an extensive experimental evaluation to verify the robustness of the proposed methods.

**Algorithm 4 ADCA**: **A**daptive **D**ata **C**orruption **A**lgorithm

---

**Input:** Covariates $X = [\mathbf{x}_1, ..., \mathbf{x}_n]$, responses $\mathbf{y} = [y_1, ..., y_n]^T$, true parameter $\mathbf{w}^*$
   penalty coefficient $\delta$, corruption index $k$, tolerance $\epsilon$
**Output:** solution $\hat{\mathbf{w}}$
1: $\mathbf{b}^0 \leftarrow \mathbf{0}, t \leftarrow 0,$
   $P_{\delta X} \leftarrow X^T(XX^T - \delta I)^{-1}X, P_\delta \leftarrow X^T(XX^T - \delta I)^{-1}\delta I$
2: **while** $\|\mathbf{b}^t - \mathbf{b}^{t-1}\|_2 > \epsilon$ **do**
3:    $\mathbf{b}^{t+1} \leftarrow HT_k(P_{\delta X}\mathbf{b}^t + (I - P_{\delta X})\mathbf{y} + P_\delta \mathbf{w}^*)$
4:    $t \leftarrow t + 1$;
5: **end while**
6: $\hat{\mathbf{w}} \leftarrow (XX^T)^{-1}X(\mathbf{y} - \mathbf{b}^t)$
7: $C \leftarrow supp(\mathbf{b}^t)$
8: **return** $\mathbf{y}_C = X_C^T \hat{\mathbf{w}}$

---

## 5.1 Data and Metrics

In our experiments, the data generation can be divided into two steps. First, we generate the basic model. The true coefficient $\mathbf{w}^*$ is chosen to be a random unit norm vector. The covariant $\mathbf{x}_i$ are iid in $\mathcal{N}(0, I_d)$. The data are generated by $y_i = \mathbf{x}_i^T w^* + \epsilon_i$, where $\epsilon_i$ are iid in $\mathcal{N}(0, \sigma^2)$. We set $\sigma = 1$ in the experiments. The second step is to generate the corrupted data using two kinds of attacks: OAA and AAA, as described in Section 5.2. The aim is to produce $k^*$ corrupted responses in the whole dataset. The prior coefficient $\mathbf{w}_0$ is generated by $\mathbf{w}^* + \nu\mathbf{u}$, where $\mathbf{u}$ is a random unit norm vector and $\nu$ is a non-negative number ($\nu$ is set to 0.5 unless otherwise stated). $\Sigma_0$ takes the form $sI$, where $s$ takes a different value for each method. All parameters are fixed in each experiment.

Following the setting in [1], we measured the performance of the regression coefficients by the standard $L_2$ error: $r_{\hat{\mathbf{w}}} = \|\hat{\mathbf{w}} - \mathbf{w}^*\|_2$. To judge whether the algorithm had converged, we used the termination criterion $\|\mathbf{w}^{t+1} - \mathbf{w}^t\|_2 \leq 10^{-4}$. All results were averaged over 10 runs.

## 5.2 Corruption Methods

To demonstrate the efficiency of our proposed methods, we apply two different attacks to the dataset: OAA and AAA. The details of these two attacks are shown as follows.

**OAA**: The set of corrupted points $S$ is selected as a uniformly random $k$-sized subset of $[n]$, and the corresponding response variables are set as $y_i = \mathbf{x}_i^T\mathbf{w}^* + b_i + \epsilon_i$, where $b_i$ are sampled from the uniform distribution $U[0, 10]$ and the white noise $\epsilon_i \sim \mathcal{N}(0, \sigma^2)$.

**AAA**: We use all information from the true data distribution to corrupt the data, and propose an adaptive data corruption algorithm (ADCA). This algorithm is quite similar to TRIP; full details of ADCA (Algorithm 4) are given in Appendix B. $\delta$ is set to $0.1n$ for $n = 1000$, $p = 200$, and to $0.2n$ for $n = 2000$, $p = 100$.

Both TRIP and BRHT employ the prior $p_{\mathbf{w}}(\mathbf{w})$ while the variance $\Sigma_0$ of $p_{\mathbf{w}}(\mathbf{w})$ in OAA is set to be four times higher than that in AAA, which means that the prior has a higher weight in AAA. Other parameters for these two algorithms are fixed. The prior distribution $p_{\mathbf{r}}(\mathbf{r})$ is set to the Gamma distribution unless otherwise stated.

We also design another leverage point attack (LPA) to test the robustness of our methods. More results can be seen in Appendix E.

## 5.3 Methods Comparison

Our methods are compared with three baselines: 1) CRR [1] is an effective robust regression method in cases where there are large numbers of random outliers; 2) Reweighted robust Bayesian regression (RRBR) [24] allows us to judge whether our proposed methods are better than the original method; 3) Rob-ULA [3] is an effective robust Bayesian inference method that approximately converges to the real posterior distribution in a finite number of steps in the presence of outliers. The parameters of RRBR are the same as in the BRHT algorithm, except for the hard thresholding part. For more comparisons of other methods, please see Appendix E.

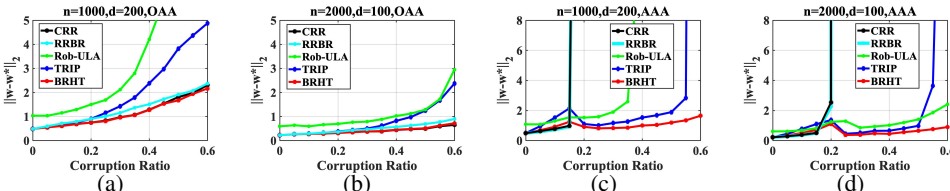

Figure 1: Recovery of parameters with respect to the number of data points $n$, dimensionality $d$, and corruption ratio $\alpha$. TRIP and BRHT are more robust under AAAs than CRR, and BRHT exhibits the best performance in all experiments. RRBR and Rob-ULA show some robustness, but offer slightly worse recovery in some experiments.

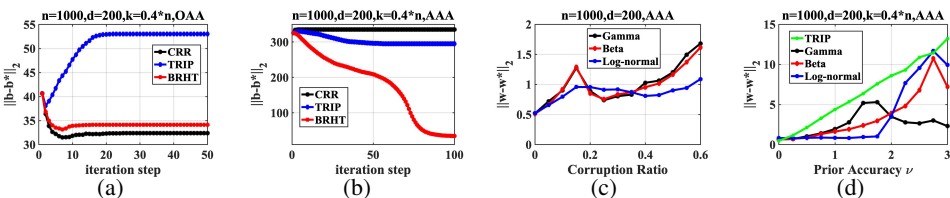

Figure 2: (a), (b) show the convergence characteristics of TRIP and BRHT. Both algorithms exhibit an estimation bias as the price of adding a prior in OAA, but BRHT is more accurate and behaves significantly better in AAA, while TRIP stalls during the iterative process. (c), (d) show the convergence under different weights of the prior $p_{\mathbf{r}}(\mathbf{r})$ and the coefficient prior $p_{\mathbf{w}}(\mathbf{w})$.

## 5.4 Recovery Properties of Coefficients and Uncorrupted Sets

CRR, RRBR, and Rob-ULA are excellent robust regression methods or Bayesian inference methods, but they all show their limitations in the face of different kinds of attacks. CRR achieves the best performance in the face of OAAs because it is theoretically unbiased, but it collapses rapidly when facing AAAs, as shown in Figures 1(c) and 1(d). RRBR and Rob-ULA take priors into consideration, but RRBR cannot resist AAAs and Rob-ULA produces poor results under OAAs, as shown in Figures 1(a)–1(d). TRIP produces a good effect against AAAs, while BRHT is not only optimal against AAAs, but also displays a similar effect to CRR in the case of OAAs. This shows that BRHT is the most robust algorithm among those compared in this experiment.

The TRIP and BRHT algorithms are compared in Figures 2(a) and 2(b). The TRIP method incorporates too much prior information in the case of OAAs, resulting in a greater estimation error than those of BRHT and CRR as shown in Figure 2(a). However, under AAAs, the prior information of TRIP and BRHT is enhanced by a factor of four, as described in Section 5.2. BRHT converges under a weaker prior, while TRIP becomes trapped around a local optimum. This shows that BRHT only needs to integrate weak prior information to ensure convergence. Figures 2(c) and 2(d) illustrate the convergence properties under different weights of the prior $p_{\mathbf{r}}(\mathbf{r})$ and coefficient prior $p_{\mathbf{w}}(\mathbf{w})$. Figure 2(c) shows that BRHT is not especially sensitive to the weight prior $p_{\mathbf{r}}(\mathbf{r})$ when this prior is relatively reliable. A log-normal distribution is the best choice when the prior $p_{\mathbf{w}}(\mathbf{w})$ is relatively close to the real parameters and a Gamma distribution is more robust when the prior is imprecise, as shown in Figure 2(d).

## 6 Conclusion

This paper has described a novel robust regression algorithm named TRIP that achieves strong results in terms of resisting AAAs. By adding a prior to the robust regression via hard thresholding, the recovery of coefficients is significantly improved. Another algorithm, named BRHT, was designed to improve the robustness of TRIP and reduce the estimation error through the use of Bayesian reweighting regression. We prove that both algorithms have strong theoretical guarantees and that the algorithms converge linearly under a mild condition. Extensive experiments have illustrated that our algorithms outperform benchmark methods in terms of both robustness and efficiency.

There are several interesting future directions to extend current work. Firstly, in this article, we only consider the case when $\mathbf{y}$ is corrupted. One would consider using the prior information to better deal with the problem where both $\mathbf{y}$ and $X$ are corrupted. Secondly, it would be also interesting to further reduce the effect of a prior on the estimation to make it consistent.

## Acknowledgments and Disclosure of Funding

This work was partly supported by National Key Research and Development Program of China (2021YFA1000300 and 2021YFA1000301), National Center for Mathematics and Interdisciplinary Sciences, and Key Laboratory of Systems and Control of CAS.

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
