# A  Details of Variational Bayesian EM Method

For the RPM model and the given covariates $X = [\mathbf{x}_1, ..., \mathbf{x}_n]$, responses $\mathbf{y} = [y_1, ..., y_n]^T$, and prior distributions $p_{\mathbf{r}}(\mathbf{r})$, $p_{\mathbf{w}}(\mathbf{w})$, the posterior of this RPM model is formulated as

$$\log p_{\mathbf{w}}(\mathbf{w}) + \log p_{\mathbf{r}}(\mathbf{r}) + \sum_{i=1}^{n} r_i \log \ell(y_i \mid \mathbf{w}, \mathbf{x}_i, \sigma^2)$$

where $\ell(y_i \mid \mathbf{w}, \mathbf{x}_i, \sigma^2)$ represent the likelihood of the normal distribution $\mathcal{N}(x_i^T \mathbf{w}, \sigma^2)$. $p_{\mathbf{w}}(\mathbf{w})$, $p_{\mathbf{r}}(\mathbf{r})$ are the priors of $\mathbf{w}$ and $\mathbf{r}$. $p(\mathbf{w})$ is the density of the normal distribution $\mathcal{N}(\mathbf{w}_0, \Sigma_0)$. We now use variational Bayesian EM to approximate the true posterior.

$$q(\mathbf{w}) \prod_i q(r_i) \approx p(\mathbf{w}, \mathbf{r}|\mathbf{y}, X, \sigma^2)$$

In the following, we derive update equations for these variational parameters.

## A.1  Derivation of $q(\mathbf{r})$ (variational E step)

Ignoring terms that do not involve $\mathbf{r}$, we take the expectations of over the remaining terms. We have

$$\log q(\mathbf{r}) = \mathbb{E}_{q(\mathbf{w})}[\log p(\mathbf{y}, \mathbf{w}, \mathbf{r}|X)] + const$$
$$= \log p_{\mathbf{r}}(\mathbf{r}) + \sum_i r_i \mathbb{E}_{q(\mathbf{w})}[\log \ell(y_i|\mathbf{w}, \mathbf{x}_i, \sigma^2)] + const$$

where $q(\mathbf{w})$ is the form of the normal distribution $\mathcal{N}(\mathbf{w}_N, V_N)$, as will be shown in the derivation of $q(\mathbf{w})$. Using this fact, we have

$$\mathbb{E}_{q(\mathbf{w})}[\log \ell(y_i|\mathbf{w}, \mathbf{x}_i, \sigma^2)] = \mathbb{E}_{q(\mathbf{w})}[-\frac{1}{2\sigma^2}(y_i - \mathbf{w}^T \mathbf{x}_i)^2 - \frac{1}{2}\log(2\pi\sigma^2)]$$
$$= -\frac{1}{2\sigma^2}[(y_i - \mathbf{w}_N^T \mathbf{x}_i)^2 + \mathbf{x}_i^T V_N \mathbf{x}_i] - \frac{1}{2}\log(2\pi\sigma^2)$$

When the priors $p_{\mathbf{r}}(r_i)$ are independent of each other, then

$$q(r_i) \propto \exp\{\log p_{\mathbf{r}}(r_i) + r_i \mathbb{E}_{q(\mathbf{w})}[\log \ell(y_i \mid \mathbf{w}, \mathbf{x}_i, \sigma^2)]\}$$

If the distribution of $q(r_i)$ is complex, we use a Markov chain Monte Carlo method to simulate the distribution. An easy example for $p_{\mathbf{r}}(r_i)$ is the Gamma distribution $Gam(r_i|a_r, b_r)$. Then,

$$q(r_i) = Gam(r_i|a_N^i, b_N^i)$$
$$a_N^i = a_r$$
$$b_N^i = b_r - \mathbb{E}_{q(\mathbf{w})}[\log \ell(y_i \mid \mathbf{w}, \mathbf{x}_i, \sigma^2)]$$
$$\mathbb{E}_{q(\mathbf{w})}[\log \ell(y_i \mid \mathbf{w}, \mathbf{x}_i, \sigma^2)] = -\frac{1}{2\sigma^2}[(y_i - \mathbf{w}_N^T \mathbf{x}_i)^2 + \mathbf{x}_i^T V_N \mathbf{x}_i] - \frac{1}{2}\log(2\pi\sigma^2)$$

for which the expectation of $r_i$ under the distribution of $q(r_i)$ can be easily obtained by $a_N^i/b_N^i$.

## A.2  Derivation of $q(\mathbf{w})$ (variational M step)

$$\log q(\mathbf{w}) = \mathbb{E}_{q(\mathbf{r})}[\log p(\mathbf{y}, \mathbf{w}, \mathbf{r}|X)] + const$$
$$= \log p_{\mathbf{w}}(\mathbf{w}) + \sum_i \mathbb{E}_{q(\mathbf{r})}(r_i) \log \ell(y_i \mid \mathbf{w}, \mathbf{x}_i, \sigma^2) + const$$
$$= -\frac{1}{2}(\mathbf{w} - \mathbf{w}_0)^T \Sigma^{-1}(\mathbf{w} - \mathbf{w}_0) - \frac{1}{2\sigma^2}(\mathbf{y} - X^T \mathbf{w})^T E_r (\mathbf{y} - X^T \mathbf{w}) + const$$
$$= -\frac{1}{2}(\mathbf{w} - \mathbf{w}_N)^T V_N^{-1}(\mathbf{w} - \mathbf{w}_N) + const$$

where $E_r$ is a matrix with diagonal entries of $E_{q(\mathbf{r})}(\mathbf{r})$ and off-diagonal elements of 0. $V_N^{-1} = \frac{1}{\sigma^2} X E_r X^T + \Sigma^{-1}$, $\mathbf{w}_N = V_N(\frac{1}{\sigma^2} X E_r \mathbf{y} + \Sigma^{-1}\mathbf{w}_0)$, and $q(\mathbf{w}) = \mathcal{N}(\mathbf{w}_N, V_N)$.

After several iterations, we use $q(\mathbf{w}) \prod_i q(r_i)$ to approximate the true posterior and take $\mathbf{w}_N$ as the MAP estimate of $q(\mathbf{w})$.

# B  Adaptive Data Corruption Method

As the original CRR algorithm can tolerate OAAs to a significant degree, we must use all information contained in the data to fool the estimator. To achieve this goal, we need to find the most suitable subset to corrupt such that the rest of the data can more likely be generated from a completely different distribution. This problem can be formulated as follows:

$$(\hat{\mathbf{w}}, \hat{S}) = \arg \min_{\substack{\mathbf{w} \in \mathbb{R}^p, S \subset [n] \\ |S| = n-k}} \sum_{i \in S} (y_i - x_i^T \mathbf{w})^2 - \delta \|\mathbf{w} - \mathbf{w}^*\|_2^2 \tag{15}$$

where $\delta$ is the penalty coefficient that determines the extent to which the parameter leaves the standard value. $\hat{S}$ is the chosen subset that cannot be corrupted. If $\delta$ is not very large, then $\sum_{i \in \hat{S}} (y_i - x_i^T \mathbf{w}^*)^2$ will be similar to $\sum_{i \in \hat{S}} (y_i - x_i^T \hat{\mathbf{w}})^2$, and so $\hat{\mathbf{w}}$ can fool the regression model into thinking that $\hat{\mathbf{w}}$ is the true parameter. From this analysis, we find that $\hat{\mathbf{w}}$ can be used to construct the corrupted data. After getting the covariates of the corrupted data, we can define the response of the corrupted data as $y_{c_i} = \mathbf{x}_{c_i}^T \hat{\mathbf{w}}$, where $\mathbf{x}_{c_i}$ is the $i^{th}$ covariate of the corrupted data.

The problem in Eq. (15) is very similar to that in Eq. (8), where $M$ is replaced by $-\delta I$ and $\mathbf{w}_0$ is replaced by $\mathbf{w}^*$. Hence, these two problems can be solved by the same method. Similar to TRIP, we proposed an adaptive data corruption algorithm (ADCA) to solve the corruption problem by replacing some parameters in TRIP. ADCA seriously destroys the data, and when the corruption ratio increases, the solution of Eq. (1) may not be close to the true parameter. However, we will see that, even in this situation, TRIP and BRHT achieve good performance, as shown in Section 5.

# C  Supplementary Material for Proofs of TRIP and BRHT Algorithms

## C.1  SSC/SSS guarantees

In this section, we introduce some theoretical properties of SSC and SSS from [2], which will be used for the convergence analysis of the proposed algorithms.

**Definition 4.** *A random variable $x \in \mathbb{R}$ is called sub-Gaussian if the following quantity is finite*

$$\sup_{p \geq 1} p^{-1/2} (E[|x|^p])^{1/p}$$

*Moreover, the smallest upper bound on this quantity is referred to as the sub-Gaussian norm of $x$ and denoted as $\|x\|_{\psi_2}$*

**Definition 5.** *A vector-valued random variable $\mathbf{x} \in \mathbb{R}^d$ is called sub-Gaussian if its unidimensional marginals $\langle \mathbf{x}, \mathbf{v} \rangle$ are sub-Gaussian for all $\mathbf{v} \in S^{d-1}$. Moreover, its sub-Gaussian norm is defined as follows*

$$\|x\|_{\psi_2} = \sup_{\mathbf{v} \in S^{d-1}} \|\langle \mathbf{x}, \mathbf{v} \rangle\|_{\psi_2}$$

**Lemma 8.** *Let $X \in \mathbb{R}^{d \times n}$ be a matrix whose columns are sampled i.i.d from a standard Gaussian distribution i.e. $\mathbf{x}_i \sim \mathcal{N}(0, I)$. Then for any $\epsilon > 0$, with probability at least $1 - \delta$, $X$ satisfies*

$$\lambda_{max}(XX^T) \leq n + (1 - 2\epsilon)^{-1} \sqrt{cnd + c'n \log \frac{2}{\delta}}$$

$$\lambda_{min}(XX^T) \geq n - (1 - 2\epsilon)^{-1} \sqrt{cnd + c'n \log \frac{2}{\delta}}$$

*where $c = 24e^2 log \frac{3}{\epsilon}$ and $c' = 24e^2$.*

**Theorem 9.** *Let $X \in \mathbb{R}^{d \times n}$ be a matrix whose columns are sampled i.i.d from a standard Gaussian distribution i.e. $\mathbf{x}_i \sim \mathcal{N}(0, I)$. Then for any $k > 0$, with probability at least $1 - \delta$, the matrix $X$ satisfies the SSC and SSS properties with constants*

$$\Lambda_k \leq k(1 + 3e\sqrt{6 \log \frac{en}{k}}) + O(\sqrt{nd + n \log \frac{1}{\delta}})$$

$$\lambda_k \geq n - (n - k)(1 + 3e\sqrt{6 \log \frac{en}{n-k}}) - \Omega(\sqrt{nd + n \log \frac{1}{\delta}})$$

**Lemma 10.** *Let $X \in \mathbb{R}^{d \times n}$ be a matrix with columns sampled from some sub-Gaussian distribution with sub-Gaussian norm $K$ and convariance $\Sigma$. Then for any $\delta > 0$, with probability at least $1 - \delta$, each of the following statements holds true:*

$$\lambda_{max}(XX^T) \leq \lambda_{max}(\Sigma) \cdot n + C_K \cdot \sqrt{dn} + t\sqrt{n}$$
$$\lambda_{min}(XX^T) \geq \lambda_{min}(\Sigma) \cdot n - C_K \cdot \sqrt{dn} - t\sqrt{n}$$

*where $t = \sqrt{\frac{1}{c_K} \log \frac{2}{\delta}}$ and $c_K$, $C_K$ are absolute constants that depend only on the sub-Gaussian norm $K$ of the distribution.*

## C.2 Convergence Proof for TRIP

**Theorem 1.** *Let $X = [\mathbf{x}_1, \ldots, \mathbf{x}_n] \in \mathbb{R}^{d \times n}$ be the given data matrix and $\mathbf{y} = X^T \mathbf{w}^* + \mathbf{b}^* + \boldsymbol{\epsilon}$ be the corrupted output with sparse corruptions of $\|\mathbf{b}^*\|_0 \leq k \cdot n$. For a specific positive semi-definite matrix $M$, the data matrix $X$ satisfies the SSC and SSS properties such that $2\frac{\Lambda_{k+k^*}}{\lambda_{min}(XX^T+M)} < 1$. Then, if $k > k^*$, it is guaranteed with a probability of at least $1 - \delta$ that, for any $\varepsilon, \delta > 0$, $\|\mathbf{b}^{T_0} - \mathbf{b}^*\|_2 \leq \varepsilon + O(e_0) + O(\frac{\sqrt{\Lambda_{k+k^*}\lambda_{max}(M)}}{\lambda_{min}(XX^T+M)})\|\mathbf{w}^* - \mathbf{w}_0\|_2$ after $T_0 = O(\log(\frac{\|\mathbf{b}^*\|_2}{\varepsilon}))$ iterations of TRIP, where $e_0 = O(\sigma\sqrt{(k + k^*)\log \frac{n}{\delta(k+k^*)}})$ under the normal design.*

*Proof.* First, we consider the iteration of the TRIP algorithm:

$$\mathbf{b}^{t+1} \leftarrow HT_k(P_{MX}\mathbf{b}^t + (I - P_{MX})y - P_{MM}\mathbf{w}_0)$$

After considering $\mathbf{y} = X^T\mathbf{w}^* + \mathbf{b}^* + \boldsymbol{\epsilon}$, the iteration step can be rewritten as:

$$\mathbf{b}^{t+1} \leftarrow HT_k(\mathbf{b}^* + X^T\boldsymbol{\lambda}^t + \mathbf{g} + \mathbf{f})$$

where

$$\boldsymbol{\lambda}^t = (XX^T + M)^{-1}X(\mathbf{b}^t - \mathbf{b}^*)$$
$$\mathbf{g} = (I - P_{MX})\boldsymbol{\epsilon}$$
$$\mathbf{f} = P_{MM}(\mathbf{w}^* - \mathbf{w}_0)$$

Because $k > k^*$, we use the property of the hard thresholding step:

$$\|\mathbf{b}_{I^{t+1}}^{t+1} - (\mathbf{b}_{I^{t+1}}^* + X_{I^{t+1}}^T\boldsymbol{\lambda}^t + \mathbf{g}_{I^{t+1}} + \mathbf{f}_{I^{t+1}})\|_2 \leq \|\mathbf{b}_{I^{t+1}}^* - (\mathbf{b}_{I^{t+1}}^* + X_{I^{t+1}}^T\boldsymbol{\lambda}^t + \mathbf{g}_{I^{t+1}} + \mathbf{f}_{I^{t+1}})\|_2$$
$$= \|X_{I^{t+1}}^T\boldsymbol{\lambda}^t + \mathbf{g}_{I^{t+1}} + \mathbf{f}_{I^{t+1}}\|_2$$

Using the trigonometric inequality:

$$\|\mathbf{b}_{I^{t+1}}^{t+1} - \mathbf{b}_{I^{t+1}}^*\|_2 \leq 2\|X_{I^{t+1}}^T\boldsymbol{\lambda}^t + \mathbf{g}_{I^{t+1}} + \mathbf{f}_{I^{t+1}}\|_2 \leq 2\|X_{I^{t+1}}^T\boldsymbol{\lambda}^t\|_2 + 2\|\mathbf{g}_{I^{t+1}}\|_2 + 2\|\mathbf{f}_{I^{t+1}}\|_2$$

Through the SSS and SSC properties of $X$, we obtain:

$$\|X_{I^{t+1}}^T\boldsymbol{\lambda}^t\|_2 = \|X_{I^{t+1}}^T(XX^T + M)^{-1}X(\mathbf{b}^{t+1} - \mathbf{b}^*)\|_2$$
$$= \|X_{I^{t+1}}^T(XX^T + M)^{-1}X_{I^t}(\mathbf{b}_{I^t}^{t+1} - \mathbf{b}_{I^t}^*)\|_2$$
$$\leq \frac{\Lambda_{k+k^*}}{\lambda_{min}(XX^T + M)}\|\mathbf{b}_{I^t}^{t+1} - \mathbf{b}_{I^t}^*\|_2$$
$$= \frac{\Lambda_{k+k^*}}{\lambda_{min}(XX^T + M)}\|\mathbf{b}^{t+1} - \mathbf{b}^*\|_2$$

According to Bhatia [1], there is a probability of at least $1 - \delta$ that, for any set $S$ of size up to $k + k^*$, we can find a uniform bound:

$$\|\boldsymbol{\epsilon}_S\|_2 \leq \sigma\sqrt{k + k^*}\sqrt{1 + 2e\sqrt{6\log \frac{en}{\delta(k + k^*)}}} \doteq e_0$$

As for $\|X\boldsymbol{\epsilon}\|_2$, Bhatia [1] gives a consistent bound of $\|X\boldsymbol{\epsilon}\|_2^2 \leq 2\sigma^2\|X\|_F^2\log(\frac{d}{\delta}) \leq 2\sigma^2 d\Lambda_n\log(\frac{d}{\delta})$, and so:

$$\|\mathbf{g}_{I^{t+1}}\|_2 = \|\boldsymbol{\epsilon}_{I^{t+1}} - X_{I^{t+1}}^T(XX^T+M)^{-1}X\boldsymbol{\epsilon}_{I^{t+1}}\|_2 \leq \|\boldsymbol{\epsilon}_{I^{t+1}}\|_2 + \|X_{I^{t+1}}^T(XX^T+M)^{-1}X\boldsymbol{\epsilon}_{I^{t+1}}\|_2$$

$$\leq e_0 + \sigma\frac{\sqrt{\Lambda_{k+k^*}\Lambda_n}}{\lambda_{min}(XX^T+M)}\sqrt{2d\log(\frac{d}{\delta})} \leq e_0 + \sigma\frac{\sqrt{\Lambda_{k+k^*}\Lambda_n}}{\lambda_n}\sqrt{2d\log(\frac{d}{\delta})}$$

$$\leq (1 + \sqrt{\frac{2d}{n}\log(\frac{d}{\delta})})e_0$$

The last inequality holds when $n$ is sufficiently large. Then, we consider $\mathbf{f}_{I^{t+1}}$:

$$\|\mathbf{f}_{I^{t+1}}\|_2 = \|X_{I^{t+1}}^T(XX^T+M)^{-1}M(\mathbf{w}^* - \mathbf{w}_0)\|_2$$

$$\leq \frac{\sqrt{\Lambda_{k+k^*}}\lambda_{max}(M)}{\lambda_{min}(XX^T+M)}\|\mathbf{w}^* - \mathbf{w}_0\|_2$$

We substitute the three calculated terms into the original result to obtain:

$$\|\mathbf{b}^{t+1} - \mathbf{b}^*\|_2 \leq 2\frac{\Lambda_{k+k^*}}{\lambda_{min}(XX^T+M)}\|\mathbf{b}^t - \mathbf{b}^*\|_2 + 2(1 + \sqrt{\frac{2d}{n}\log(\frac{d}{\delta})})e_0$$

$$+ 2\frac{\sqrt{\Lambda_{k+k^*}}\lambda_{max}(M)}{\lambda_{min}(XX^T+M)}\|\mathbf{w}^* - \mathbf{w}_0\|_2$$

We let $\eta = 2\frac{\Lambda_{k+k^*}}{\lambda_{min}(XX^T+M)}$. Because $\mathbf{b}^0 = 0$:

$$\|\mathbf{b}^{t+1} - \mathbf{b}^*\|_2 \leq \eta^t\|\mathbf{b}^*\|_2 + \frac{2}{1-\eta}(1 + \sqrt{\frac{2d}{n}\log(\frac{d}{\delta})})e_0$$

$$+ \frac{2}{1-\eta}\frac{\sqrt{\Lambda_{k+k^*}}\lambda_{max}(M)}{\lambda_{min}(XX^T+M)}\|\mathbf{w}^* - \mathbf{w}_0\|_2$$

Suppose that $n > d\log(d)$. Then, $1 + \sqrt{\frac{2d}{n}\log(\frac{d}{\delta})} = O(1)$. From the expression of $e_0$, we have that $e_0 = O(\sigma\sqrt{(k+k^*)\log\frac{n}{\delta(k+k^*)}})$. Then, after $T_0 = O(\log(\frac{\|\mathbf{b}^*\|_2}{\varepsilon}))$, we obtain:

$$\|\mathbf{b}^{T_0} - \mathbf{b}^*\|_2 \leq \varepsilon + O(e_0) + O(\frac{\sqrt{\Lambda_{k+k^*}}\lambda_{max}(M)}{\lambda_{min}(XX^T+M)})\|\mathbf{w}^* - \mathbf{w}_0\|_2$$

$\square$

**Theorem 2.** *Under the conditions of Theorem 1 and assuming that $\mathbf{x}_i \in \mathbb{R}^d$ are generated from the standard normal distribution, for $k > k^*$, it is guaranteed with a probability of at least $1 - \delta$ that, for any $\varepsilon, \delta > 0$, the current estimation coefficient $\mathbf{w}_{T_0}$ satisfies $\|\mathbf{w}_{T_0} - \mathbf{w}^*\|_2 \leq O(\frac{1}{\sqrt{n}})(\varepsilon + e_0) + O(\frac{\sqrt{k+k^*}\lambda_{max}(M)}{n^{3/2}})\|\mathbf{w}^* - \mathbf{w}_0\|_2$ after $T_0 = O(\log(\frac{\|\mathbf{b}^*\|_2}{\varepsilon}))$ steps.*

*Proof.*

$$\mathbf{w}^t = (XX^T)^{-1}X(\mathbf{y}-\mathbf{b}^t) = (XX^T)^{-1}X(X^T\mathbf{w}^*+\mathbf{b}^*+\boldsymbol{\epsilon}-\mathbf{b}^t) = \mathbf{w}^*+(XX^T)^{-1}X(\boldsymbol{\epsilon}+\mathbf{b}^*-\mathbf{b}^t)$$

$$\|\mathbf{w}^t - \mathbf{w}^*\|_2 = \|(XX^T)^{-1}X(\boldsymbol{\epsilon} + \mathbf{b}^* - \mathbf{b}^t)\|_2 \leq \frac{1}{\lambda_n}(\|X\boldsymbol{\epsilon}\|_2 + \|X(\mathbf{b}^* - \mathbf{b}^t)\|_2)$$

$$\leq \frac{\sqrt{\Lambda_n}}{\lambda_n}\sigma\sqrt{2d\log(\frac{d}{\delta})} + \frac{1}{\lambda_n}\|X(\mathbf{b}^* - \mathbf{b}^t)\|_2)$$

$$\leq \frac{\sqrt{\Lambda_n}}{\lambda_n}\sigma\sqrt{2d\log(\frac{d}{\delta})} + \frac{\sqrt{\Lambda_n}}{\lambda_n}\left[\eta^t\|\mathbf{b}^*\|_2 + \frac{2}{1-\eta}(1 + \sqrt{\frac{2d}{n}\log(\frac{d}{\delta})})e_0\right.$$

$$\left.+ \frac{2}{1-\eta}\frac{\sqrt{\Lambda_{k+k^*}}\lambda_{max}(M)}{\lambda_{min}(XX^T+M)}\|\mathbf{w}^* - \mathbf{w}_0\|_2\right]$$

when $n$ is sufficiently large. By Lemma 8 and Theorem 9, $\sqrt{\Lambda_n}/\lambda_n$ can then be approximated as $O(1/\sqrt{n})$ and $\sqrt{\Lambda_{k+k^*}}$ can be approximated as $O(\sqrt{k+k^*})$. Then, we have:

$$\|\mathbf{w}_t - \mathbf{w}^*\|_2 \leq O(\frac{1}{\sqrt{n}})(\varepsilon + e_0) + O(\frac{\sqrt{k+k^*}\lambda_{\max}(M)}{n^{3/2}})\|\mathbf{w}^* - \mathbf{w}_0\|_2$$

$\square$

**Theorem 3.** *Let* $X = [\mathbf{x}_1, ..., \mathbf{x}_n] \in \mathbb{R}^{d \times n}$ *be the given matrix with each* $\mathbf{x}_i \sim \mathcal{N}(\mathbf{0}, \Sigma)$. *Let* $\mathbf{y} = X^T\mathbf{w}^* + \mathbf{b} + \boldsymbol{\epsilon}$ *and* $\|\mathbf{b}\|_0 \leq k^*$. *Also, let* $k^* \leq k$ *and suppose* $\lim_{n\to\infty} \frac{\lambda_{min}(M)}{n} = \xi$. *Then if the following equation holds*

$$2\frac{k+k^*}{n}(1 + 3e\sqrt{6\log\frac{en}{k+k^*}}) < 1 + \xi$$

*and* $n \geq \Omega(d + \log\frac{1}{\delta})$. *Then, with probability at least* $1 - \delta$, *the data satisfies* $2\frac{\Lambda_{k+k^*}}{\lambda_{min}(XX^T+M)} < 1$, *More specifically, after* $T_0 = O(\log(\frac{\|\mathbf{b}^*\|_2}{\varepsilon}))$ *steps in TRIP algorithm, the estimation coefficient* $\mathbf{w}_{T_0}$ *satisfies* $\|\mathbf{w}_{T_0} - \mathbf{w}^*\|_2 \leq O(\frac{1}{\sqrt{n}})(\varepsilon + e_0) + O(\frac{\sqrt{k+k^*}\lambda_{\max}(M)}{n^{3/2}})\|\mathbf{w}^* - \mathbf{w}_0\|_2$.

*Proof.* We notice that if $\mathbf{x} \sim \mathcal{N}(\mathbf{0}, \Sigma)$, then $\Sigma^{-1/2}\mathbf{x} \sim \mathcal{N}(\mathbf{0}, I)$. Thus by Theorem 9 and Lemma 8, with the probability at least $1 - \delta$, the data matrix $\tilde{X} = \Sigma^{1/2}X$ satisfies SSC and SSS properties with the following constants

$$\Lambda_k \leq k(1 + 3e\sqrt{6\log\frac{en}{k}}) + O(\sqrt{nd + n\log\frac{1}{\delta}}),$$

$$\lambda_{min}(XX^T) \geq n - (1-2\epsilon)^{-1}\sqrt{cnd + c'n\log\frac{2}{\delta}}.$$

As seen in Theorem 1, the convergence of TRIP needs to satisfies $2\frac{\Lambda_{k+k^*}}{\lambda_{min}(XX^T+M)} < 1$. We notice that $\lambda_{min}(XX^T + M) \geq \lambda_{min}(XX^T) + \lambda_{min}(M)$, so the convergence condition can be scaled to $2\Lambda_{k+k^*} \leq \lambda_{min}(XX^T) + \lambda_{min}(M)$. Using the above bounds, the condition is translated into

$$\underbrace{2\frac{k+k^*}{n}(1 + 3e\sqrt{6\log\frac{en}{k+k^*}})}_{(A)} + \underbrace{O(\sqrt{\frac{d}{n} + \frac{1}{n}\log\frac{1}{\delta}})}_{(B)} < 1 + \frac{\lambda_{min}(M)}{n}.$$

For $n = \Omega(d + \frac{1}{\delta})$ and suppose $n$ is large enough, the part $(B)$ goes to 0. Also because $\lim_{n\to\infty} \frac{\lambda_{min}(M)}{n} = \xi$, so the condition becomes

$$2\frac{k+k^*}{n}(1 + 3e\sqrt{6\log\frac{en}{k+k^*}}) < 1 + \xi.$$

$\square$

The condition $2\frac{k+k^*}{n}(1 + 3e\sqrt{6\log\frac{en}{k+k^*}}) < 1 + \xi$ seems quite abstract. By approximating $f(t) = 2t(1 + 3e\sqrt{6\log\frac{e}{t}})$ using its second order Taylor's expansion at $t = 1/10$, which is shown in Figure 3. We can give an approximated breakdown point of TRIP algorithm when $\xi$ is not too large, i.e.,

$$k^* \leq k \leq (0.3023 - \sqrt{0.0887 - 0.0040\xi})n. \tag{16}$$

### C.3 Convergence Proof for BRHT

### C.3.1 Proof of Theorem 5

**Lemma 4.** *For any real function* $f(x)$:

$$\sup_{x\geq 0}[f(x) + ax] \leq \sup_{x\geq 0}[f(x) + bx] \tag{17}$$

*for any* $b \geq a \geq 0$.

*Proof.* Suppose the lemma does not hold, that is, $a \geq 0, b \geq a$, but

$$\sup_{x \geq 0}[f(x) + ax] > \sup_{x \geq 0}[f(x) + bx]$$

We select the array $\{x_n\} = \{x_1, x_2, \dots\}$ such that $\lim_{i \to \infty}[f(x_i) + ax_i] = \sup_{x \geq 0}[f(x) + ax]$. Then, we consider the set $S \doteq \{f(x_i) + bx_i | x_i \in \{x_n\}\}$. It is easy to see that $\sup S \geq \sup_{x \geq 0}[f(x) + ax]$ and $\sup S \leq \sup_{x \geq 0}[f(x) + bx]$. As shown above, however, $\sup_{x \geq 0}[f(x) + ax] > \sup_{x \geq 0}[f(x) + bx]$. This is a contradiction, and Lemma 4 is proved. $\square$

**Theorem 5.** *Suppose the prior of $r_i$ is independently and identically distributed (iid). We consider the $t^{th}$ iteration step of the BRHT algorithm, in which $\mathbf{w}_t, \mathbf{r}_t = \arg\max_{\mathbf{w} \in \mathbb{R}^d, \mathbf{r} \in \mathbb{R}_+^n} M(\mathbf{w}, \mathbf{r}, \mathbf{b}_t)$, where $\mathbf{b}_t = HT_k(\mathbf{y} - X^T\mathbf{w}_{t-1})$ is obtained from the hard thresholding step. Then, we have that $U(\mathbf{w}_t, \mathbf{r}_t, S_{t+1}) \geq U(\mathbf{w}_{t-1}, \mathbf{r}_{t-1}, S_t)$.*

*Proof.* After obtaining $\mathbf{b}_t$ by $\mathbf{b}_t = HT_k(\mathbf{y} - X^T\mathbf{w}_{t-1})$, we consider $M(\mathbf{w}_{t-1}, \mathbf{r}_{t-1}, \mathbf{b}_t)$, that is:

$$M(\mathbf{w}_{t-1}, \mathbf{r}_{t-1}, \mathbf{b}_t) = \log p_{\mathbf{w}}(\mathbf{w}_{t-1}) + \sum_{i \in S_t}[\log p_{\mathbf{r}}(r_i^{t-1}) + r_i^{t-1} \log \ell(y_i \mid \mathbf{w}_{t-1}, \mathbf{x}_i, \sigma^2)]$$

$$+ \sum_{j \in [n] \setminus S_t}[p_{\mathbf{r}}(r_j^{t-1}) + r_j^{t-1}\ell(0)]$$

where $\ell(0)$ is the value of the likelihood of $\mathcal{N}(0, \sigma^2)$. This is because, after the hard thresholding step and if $i$ is not chosen from the clean set,' $y_i - b_i^t = y_i - (y_i - X^T\mathbf{w}_{t-1}) = X^T\mathbf{w}_{t-1}$. Thus, it can be seen that $\ell(y_i - b_i^t \mid \mathbf{w}_{t-1}, \mathbf{x}_i, \sigma^2) = \ell(\mathbf{x}_i^T\mathbf{w}_{t-1} \mid \mathbf{w}_{t-1}, \mathbf{x}_i, \sigma^2) = \ell(0)$. We consider a pseudo-reweighting process (this is just for the convenience of the proof and does not appear in the algorithm, but does not affect the result of the algorithm). We try to maximize $M(\mathbf{w}_{t-1}, \mathbf{r}, \mathbf{b}_t)$ by varying $\mathbf{r}$. Because of the independence of $p_{\mathbf{r}}(r_i)$ and the definition of $M(\mathbf{w}_{t-1}, \mathbf{r}_{t-1}, \mathbf{b}_t)$, the value of $\mathbf{r}$ in $S_t$ is unchanged.

$$\tilde{M}_{t-1} = \max_{\mathbf{r} \in \mathbb{R}^n} M(\mathbf{w}_{t-1}, \mathbf{r}, \mathbf{b}_t)$$

$$= \log p_{\mathbf{w}}(\mathbf{w}_{t-1}) + \sum_{i \in S_t}[\log p_{\mathbf{r}}(r_i^{t-1}) + r_i^{t-1} \log \ell(y_i \mid \mathbf{w}_{t-1}, \mathbf{x}_i, \sigma^2)] + kg(0)$$

$$= U(\mathbf{w}_{t-1}, \mathbf{r}_{t-1}, S_t) + kg(0)$$

where $g(0)$ is defined as $\max_{r_i}[p_{\mathbf{r}}(r_i) + r_i\ell(0)]$. Next, we consider the update of $\mathbf{w}$. Because:

$$\mathbf{w}_t, \mathbf{r}_t = \arg\max_{\mathbf{w} \in \mathbb{R}^d, \mathbf{r} \in \mathbb{R}_+^n} M(\mathbf{w}, \mathbf{r}, \mathbf{b}_t)$$

it is easy to see that:

$$M(\mathbf{w}_t, \mathbf{r}_t, \mathbf{b}_t) \geq \max_{\mathbf{r} \in \mathbb{R}_+^n} M(\mathbf{w}_{t-1}, \mathbf{r}, \mathbf{b}_t) = \tilde{M}_{t-1}$$

Finally, we examine $\tilde{M}_t = \max_{\mathbf{r} \in \mathbb{R}^n} M(\mathbf{w}_t, \mathbf{r}, \mathbf{b}_{t+1})$. The explicit form of $\tilde{M}_t$ can be given by $\tilde{M}_{t-1}$. We compare the $\tilde{M}_t$ and $M(\mathbf{w}_t, \mathbf{r}_t, \mathbf{b}_t)$:

$$\tilde{M}_t - M(\mathbf{w}_t, \mathbf{r}_t, \mathbf{b}_t) = \log p_{\mathbf{w}}(\mathbf{w}_t) + \sum_{i \in S_{t+1}}[\log p_{\mathbf{r}}(r_i^t) + r_i^t \log \ell(y_i \mid \mathbf{w}_t, \mathbf{x}_i, \sigma^2)] + kg(0)$$

$$- \{\log p_{\mathbf{w}}(\mathbf{w}_t) + \sum_{j \in S_t}[\log p_{\mathbf{r}}(r_j^t) + r_j^t \log \ell(y_j \mid \mathbf{w}_t, \mathbf{x}_j, \sigma^2)]$$

$$+ \sum_{j \in [n] \setminus S_t}[p_{\mathbf{r}}(r_j^t) + r_j^t \log \ell(y_j - b_j^t \mid \mathbf{w}_t, \mathbf{x}_j, \sigma^2)]\}$$

$$= \sum_{i \in S_{t+1} \setminus S_t}[\log p_{\mathbf{r}}(r_i^t) + r_i^t \log \ell(y_i \mid \mathbf{w}_t, \mathbf{x}_i, \sigma^2)] - \sum_{j \in S_t \setminus S_{t+1}}[\log p_{\mathbf{r}}(r_j^t) + r_j^t \log \ell(y_j \mid \mathbf{w}_t, \mathbf{x}_j, \sigma^2)]$$

$$+ kg(0) - \sum_{j \in [n] \setminus S_t}[p_{\mathbf{r}}(r_j^t) + r_j^t \log \ell(y_j - b_j^t \mid \mathbf{w}_t, \mathbf{x}_j, \sigma^2)]$$

Following the hard thresholding step, $\forall i \in S_{t+1} \backslash S_t$ and $\forall j \in S_t \backslash S_{t+1}$, $|y_i - \mathbf{x}_i^T \mathbf{w}_t| \le |y_j - \mathbf{x}_j^T \mathbf{w}_t|$, and so $\log \ell(y_i \mid \mathbf{w}_t, \mathbf{x}_i, \sigma^2) \ge \log \ell(y_j \mid \mathbf{w}_t, \mathbf{x}_j, \sigma^2)$. By Lemma 4, we have that:

$$\log p_{\mathbf{r}}(r_i^t) + r_i^t \log \ell(y_i \mid \mathbf{w}_t, \mathbf{x}_i, \sigma^2) \ge \log p_{\mathbf{r}}(r_j^t) + r_j^t \log \ell(y_j \mid \mathbf{w}_t, \mathbf{x}_j, \sigma^2)$$

and because $\forall j \in [n] \backslash S_t$, $\log \ell(y_j - b_j^t \mid \mathbf{w}_t, \mathbf{x}_j, \sigma^2) \le \ell(0)$, we have:

$$\log p_{\mathbf{r}}(r_j^t) + r_j^t \log \ell(y_j - b_j^t \mid \mathbf{w}_t, \mathbf{x}_j, \sigma^2) \le g(0)$$

This proves that:

$$\tilde{M}_t \ge M(\mathbf{w}_t, \mathbf{r}_t, \mathbf{b}_t)$$

Note that $M(\mathbf{w}_t, \mathbf{r}_t, \mathbf{b}_t) \ge \tilde{M}_{t-1}$. Using the expressions for $\tilde{M}_t$ and $\tilde{M}_{t-1}$:

$$U(\mathbf{w}_t, \mathbf{r}_t, S_{t+1}) + kg(0) \ge U(\mathbf{w}_{t-1}, \mathbf{r}_{t-1}, S_t) + kg(0)$$
$$U(\mathbf{w}_t, \mathbf{r}_t, S_{t+1}) \ge U(\mathbf{w}_{t-1}, \mathbf{r}_{t-1}, S_t)$$

$\square$

### C.3.2 Proof of Theorems 6 and 7

To prove Theorem 4, we require a certain assumption. We will show that this assumption is reasonable through a brief description in Appendix C.2.3.

**Assumption 1.** *Let $X$ be the given data matrix and $\mathbf{y} = X^T \mathbf{w}^* + \mathbf{b}^* + \epsilon$ be the output. For any specific positive semi-definite matrix $M$, there exist $\alpha > 0$ and $0 < \gamma \le 1 + \epsilon$, where $\epsilon$ is a small positive number, that for any estimation $\hat{\mathbf{b}}$ of $\mathbf{b}^*$, and let $I_{\hat{\mathbf{b}}} = supp(\hat{\mathbf{b}}) \cup supp(\mathbf{b}^*)$, it holds that*

$$u_1 = \|\epsilon_{I_{\hat{\mathbf{b}}}} + X_{I_{\hat{\mathbf{b}}}}^T(\mathbf{w}^* - \mathbf{w}_1)\|_2 \le \gamma \|\epsilon_{I_{\hat{\mathbf{b}}}} + X_{I_{\hat{\mathbf{b}}}}^T(\mathbf{w}^* - \mathbf{w}_2)\|_2 = \gamma u_2$$

*where $\mathbf{w}_1$ and $\mathbf{w}_2$ are obtained from:*

$$\mathbf{w}_1 = VBEM(X, \mathbf{y} - \hat{\mathbf{b}}, p_{\mathbf{r}}(\mathbf{r}), p_{\mathbf{w}}(\mathbf{w}))$$

$$\mathbf{w}_2 = \arg \min_{\mathbf{w} \in \mathbb{R}^d} \sum_{i=1}^n \|y_i - \hat{b}_i - \mathbf{x}_i^T \mathbf{w}\|^2 + (\mathbf{w} - \mathbf{w}_0)^T M(\mathbf{w} - \mathbf{w}_0)$$

*and $p_{\mathbf{w}}(\mathbf{w}) = \mathcal{N}(\mathbf{w}_0, \alpha\sigma^2 M^{-1})$*

This assumption can be easily understood as making the Bayesian reweighting regression more robust and accurate than simple regression, thus providing a more reliable solution in each iteration step of the BRHT algorithm. This can be explained from the following two aspects: 1) the Bayesian reweighting regression adds smaller weights to points with large deviations, so the regression is less affected by outliers, especially when the estimation $\hat{\mathbf{b}}$ is not very accurate. 2) By considering the robustness of the Bayesian reweighting regression, smaller prior weights are required to recover the true coefficient. Thus, $\mathbf{w}_2$ is closer to the true coefficient $\mathbf{w}^*$ than $\mathbf{w}_1$, which is reflected in the prior shrinkage coefficient $\alpha$ and the error shrinkage coefficient $\gamma$.

**Theorem 6.** *Consider a data matrix $X$ and a specific positive semi-definite matrix $M$ satisfying the SSC and SSS properties such that $2\frac{\Lambda_{k+k^*}}{\lambda_{min}(XX^T+M)} < 1$. Then, there exist $\alpha > 0$ and $0 < \gamma \le 1 + \epsilon$, where $\epsilon$ is a small number, such that if $k > k^*$ and $\Sigma_0$ in the prior $p_{\mathbf{w}}(\mathbf{w})$ is $\alpha\sigma^2 M^{-1}$, it is guaranteed with a probability of at least $1 - \delta$ that, for any $\varepsilon, \delta > 0$, $\|\mathbf{b}^{T_0} - \mathbf{b}^*\|_2 \le \varepsilon + O(e_0) + O(\frac{\sqrt{\Lambda_{k+k^*}\lambda_{max}(M)}}{\lambda_{min}(XX^T+M)})\gamma\|\mathbf{w}^* - \mathbf{w}_0\|_2$ after $T_0 = O(\log(\frac{\gamma\|\mathbf{b}^*\|_2}{\varepsilon}))$ iterations of BRHT, where $e_0 = O(\sigma\sqrt{(k+k^*)\log\frac{n}{\delta(k+k^*)}})$ under the normal design.*

*Proof.* The iteration step of the BRHT algorithm is:

$$\mathbf{b}^{t+1} \leftarrow HT_k(\mathbf{y} - X^T \mathbf{w}^t)$$

where $\mathbf{w}^t = VBEM(X, \mathbf{y} - \mathbf{b}^t, p_{\mathbf{r}}(\mathbf{r}), p_{\mathbf{w}}(\mathbf{w}))$ and:

$$\|\mathbf{b}_{I^{t+1}}^{t+1} - (\mathbf{y}_{I^{t+1}} - X_{I^{t+1}}^T \mathbf{w}^t)\|_2 \leq \|\mathbf{b}_{I^{t+1}}^* - (\mathbf{y}_{I^{t+1}} - X_{I^{t+1}}^T \mathbf{w}^t)\|_2$$
$$= \|\mathbf{b}_{I^{t+1}}^* - (\mathbf{b}_{I^{t+1}}^* + \epsilon_{I_{t+1}} + X_{I_{t+1}}^T (\mathbf{w}^* - \mathbf{w}^t))\|_2$$
$$= \|\epsilon_{I_{t+1}} + X_{I_{t+1}}^T (\mathbf{w}^* - \mathbf{w}^t)\|_2$$

By defining $\hat{\mathbf{w}}^t = (XX^T + M)^{-1}(X(\mathbf{y} - \mathbf{b}^t) + M\mathbf{w}_0)$ and using the trigonometric inequality, we obtain:

$$\|\mathbf{b}_{I^{t+1}}^{t+1} - \mathbf{b}_{I^{t+1}}^*\|_2 \leq 2\|\epsilon_{I_{t+1}} + X_{I_{t+1}}^T (\mathbf{w}^* - \mathbf{w}^t)\|_2$$
$$\leq 2\gamma\|\epsilon_{I_{t+1}} + X_{I_{t+1}}^T (\mathbf{w}^* - \hat{\mathbf{w}}^t)\|_2$$
$$= 2\gamma\|X_{I^{t+1}}^T \lambda^t + \mathbf{g}_{I^{t+1}} + \mathbf{f}_{I^{t+1}}\|_2$$

The second inequality holds because of assumption 1. $\lambda^t, \mathbf{g}, \mathbf{f}$ have the same meaning as in Theorem 1. Therefore, through the same proof procedure as for Theorem 1, the above inequality can be finally transformed into the following formula:

$$\|\mathbf{b}^{t+1} - \mathbf{b}^*\|_2 \leq 2\gamma \frac{\Lambda_{k+k^*}}{\lambda_{min}(XX^T + M)} \|\mathbf{b}^t - \mathbf{b}^*\|_2 + 2\gamma(1 + \sqrt{\frac{2d}{n} \log(\frac{d}{\delta})})e_0$$
$$+ 2\gamma \frac{\sqrt{\Lambda_{k+k^*}}\lambda_{max}(M)}{\lambda_{min}(XX^T + M)} \|\mathbf{w}^* - \mathbf{w}_0\|_2$$

We let $\eta = 2\gamma \frac{\Lambda_{k+k^*}}{\lambda_{min}(XX^T + M)}$, and because $\mathbf{b}^0 = 0$, we can write:

$$\|\mathbf{b}^{t+1} - \mathbf{b}^*\|_2 \leq \eta^t \|\mathbf{b}^*\|_2 + \frac{2\gamma}{1 - \eta}(1 + \sqrt{\frac{2d}{n} \log(\frac{d}{\delta})})e_0$$
$$+ \frac{2\gamma}{1 - \eta} \frac{\sqrt{\Lambda_{k+k^*}}\lambda_{max}(M)}{\lambda_{min}(XX^T + M)} \|\mathbf{w}^* - \mathbf{w}_0\|_2$$

Suppose that $n > d\log(d)$. Then, $1 + \sqrt{\frac{2d}{n} \log(\frac{d}{\delta})} = O(1)$. From the expression for $e_0$, we have that $e_0 = O(\sigma\sqrt{(k+k^*)\log\frac{n}{\delta(k+k^*)}})$. Then, after $T_0 = O(\log(\frac{\|\mathbf{b}^*\|_2}{\varepsilon}))$, we have:

$$\|\mathbf{b}^{T_0} - \mathbf{b}^*\|_2 \leq \varepsilon + O(e_0) + O(\frac{\sqrt{\Lambda_{k+k^*}}\lambda_{max}(M)}{\lambda_{min}(XX^T + M)})\gamma\|\mathbf{w}^* - \mathbf{w}_0\|_2$$

$\square$

**Theorem 7.** *Under the conditions of Theorem 4 and assuming that $\mathbf{x}_i \in \mathbb{R}^d$ are generated from the standard normal distribution, there exist $\alpha > 0$ and $0 < \gamma \leq 1 + \epsilon$, where $\epsilon$ is a small number, such that if $k > k^*$ and $\Sigma_0$ in the prior $p_{\mathbf{w}}(\mathbf{w})$ is $\alpha\sigma^2 M^{-1}$, it is guaranteed with a probability of at least $1 - \delta$ that, for any $\varepsilon, \delta > 0$, the current estimation coefficient $\mathbf{w}_{T_0}$ satisfies $\|\mathbf{w}_{T_0} - \mathbf{w}^*\|_2 \leq O(\frac{1}{\sqrt{n}})(\varepsilon + e_0) + O(\frac{\sqrt{k+k^*}\lambda_{\max}(M)}{n^{3/2}})\gamma\|\mathbf{w}^* - \mathbf{w}_0\|_2$ after $T_0 = O(\log(\frac{\gamma\|\mathbf{b}^*\|_2}{\varepsilon}))$ steps.*

The proof of Theorem 7 is the same as that for Theorem 2, so it is omitted here.

### C.3.3 Rationality of Assumption 1

In this section, we use some simulations to check whether assumption 1 is true in the iteration of the BRHT algorithm. For this problem, we choose some special $M$ under AAA to make our description more representative. For each corruption ratio, we choose $M$ so as to achieve the minimum fitting error $\|\mathbf{w}_t - \mathbf{w}^*\|_2$ in the TRIP algorithm. We then find a prior shrinkage coefficient $\alpha$ for this $M$ and simulate the BRHT algorithm to show that there exists an error shrinkage coefficient $\gamma$ such that the following formula holds in all iterative steps:

$$u_{1t} = \|\epsilon_{I_t} + X_{I_t}^T (\mathbf{w}^* - \mathbf{w}_{1t})\|_2 \leq \gamma\|\epsilon_{I_t} + X_{I_t}^T (\mathbf{w}^* - \mathbf{w}_{2t})\|_2 = \gamma u_{2t}$$

| Corruption rate | $\gamma$ |
|:---:|:---:|
| 0.2 | 1.007 |
| 0.25 | 0.937 |
| 0.3 | 0.917 |
| 0.35 | 0.876 |
| 0.4 | 0.879 |
| 0.45 | 0.958 |
| 0.5 | 1.000 |
| 0.55 | 1.008 |

Table 1: Calculated $\gamma$ for each corruption rate

| Corruption rate | mean($u_{1t}/u_{2t}$) |
|:---:|:---:|
| 0.2 | 0.989 |
| 0.25 | 0.917 |
| 0.3 | 0.892 |
| 0.35 | 0.857 |
| 0.4 | 0.806 |
| 0.45 | 0.827 |
| 0.5 | 0.772 |
| 0.55 | 0.758 |

Table 2: Average error ratio for each corruption rate

where $\mathbf{w}_{1t}$ and $\mathbf{w}_{2t}$ are obtained from:

$$\mathbf{w}_{1t} = VBEM(X, \mathbf{y} - \mathbf{b}^t, p_{\mathbf{r}}(\mathbf{r}), p_{\mathbf{w}}(\mathbf{w}))$$

$$\mathbf{w}_{2t} = \arg\min_{\mathbf{w} \in \mathbb{R}^d} \sum_{i=1}^{n} \|y_i - b_i^t - \mathbf{x}_i^T \mathbf{w}\|^2 + (\mathbf{w} - \mathbf{w}_0)^T M (\mathbf{w} - \mathbf{w}_0)$$

and $p_{\mathbf{w}}(\mathbf{w}) = \mathcal{N}(\mathbf{w}_0, \alpha\sigma^2 M^{-1})$. For the chosen corruption ratio, we take eight evenly spaced points from 0.2 to 0.55. This is because the CRR method collapses when the corruption ratio exceeds 0.2, so including the prior become very important at this time. By selecting an appropriate prior shrinkage coefficient $\alpha$ for different $M$, the overall result is as shown in Figure 4.

We can find $\gamma$ for each $M$ by calculating $\max(u_{1t}/u_{2t})$, where $t = 1, ...$ are the iterative steps until convergence. The results are presented in Table 1. For most corruption rates, $\gamma < 1$. However, there are still some cases where $\gamma$ is greater than 1. This is because, in the iteration process, there are few steps in which $u_{1t}$ and $u_{2t}$ are very close, while in most cases they are well separated. We use another criterion, $mean(u_{1t}/u_{2t})$, to show this phenomenon; the results are presented in Table 2. We can see that, as the corruption ratio increases, $mean(u_{1t}/u_{2t})$ basically exhibits a downward trend and all values are less than 1. This indicates that the performance of BRHT is usually better than that suggested by the theory. This experiment can be used to explain why BRHT usually outperforms TRIP, even when TRIP uses the optimal parameters.

## D  Choice of Hyperparameters in BRHT

In the BRHT algorithm, the most important parameter for model performance is the parameter in the weight prior $p_{\mathbf{r}}(\mathbf{r})$. According to assumption 1, we must ensure that the Bayesian reweighting regression provides a more robust and accurate solution than traditional least-squares regression. This requires the weight $E_{q(\mathbf{r})}(\mathbf{r})$ in the variational M step of the VBEM algorithm to be relatively insensitive to $\mathbb{E}_{q(\mathbf{w})}[\log \ell(y_i \mid \mathbf{w}, \mathbf{x}_i, \sigma^2)]$, or very few points will have large weights and others will have little impact on the estimates. A relatively sensitive weight will lead to bias, as only a few points of information will be used, and the effects will be even worse when some outliers have not

been detected. Additionally, the weight cannot be too stable, or BRHT will have almost the same performance as TRIP. Here, we present a useful way to determine the hyperparameters so that all uncorrupted points have relatively large weights when the regression result is correct.

Consider the variational E step in the VBEM method. We have:

$$q(r_i) \propto exp\{\log p_{\mathbf{r}}(r_i) + r_i \mathbb{E}_{q(\mathbf{w})}[\log \ell(y_i \mid \mathbf{w}, \mathbf{x}_i, \sigma^2)]\}$$

We use the true likelihood $\log \ell(y_i \mid \mathbf{w}^*, \mathbf{x}_i, \sigma^2)$ to replace $\mathbb{E}_{q(\mathbf{w})}[\log \ell(y_i \mid \mathbf{w}, \mathbf{x}_i, \sigma^2)]$, and we find that:

$$\log \ell(y_i \mid \mathbf{w}^*, \mathbf{x}_i, \sigma^2) = -\frac{1}{2\sigma^2}(y_i - \mathbf{x}_i^T \mathbf{w}^*)^2 - \frac{1}{2}\log(2\pi\sigma^2)$$

If $y_i$ is not corrupted, then $\frac{1}{\sigma^2}(y_i - \mathbf{x}_i^T \mathbf{w}^*)^2 = \frac{1}{\sigma^2}\epsilon_i^2 \leq \chi^2(0.95)$ holds with at least 95% probability, where $\chi^2(0.95)$ is the 95% quantile of the $\chi^2$ distribution with 1 degree of freedom. Under the above condition, with at least 95% probability, it is easy to see that:

$$-\frac{1}{2}\chi^2(0.95) - \frac{1}{2}\log(2\pi\sigma^2) \leq \log \ell(y_i \mid \mathbf{w}^*, \mathbf{x}_i, \sigma^2) \leq -\frac{1}{2}\log(2\pi\sigma^2)$$

Here, we define two posterior distributions of weights in the extreme case where all points fit well or deviate greatly in the true regression model:

$$q_1(r_i) \propto exp[\log p_{\mathbf{r}}(r_i) + r_i(-\frac{1}{2}\log(2\pi\sigma^2))]$$

$$q_2(r_i) \propto exp[\log p_{\mathbf{r}}(r_i) + r_i(-\frac{1}{2}\chi^2(0.95) - \frac{1}{2}\log(2\pi\sigma^2))]$$

Then, the hyperparameter in the weight prior $p_{\mathbf{r}}(\mathbf{r})$ is determined by the following rule:

$$\mathbb{E}_{q_2(r_i)}(r_i) \geq \beta \mathbb{E}_{q_1(r_i)}(r_i)$$

In this paper, $\beta = \frac{1}{2}$. The parameter $\sigma^2$ can be replaced by a robust estimate such as the M estimator. After the weight prior $p_{\mathbf{r}}(\mathbf{r})$ has been determined, the hyperparameter $\Sigma$ in prior $p_{\mathbf{w}}(\mathbf{w})$ can be selected by cross-validation using $\Sigma$ in the specific form $\Sigma = sI$.

# E    Additional Experimental Results

In this section, we give more experimental results of TRIP and BRHT in comparison with alternative methods. We also show the robustness of our methods under other attacks. First, We compare TRIP and BRHT with the TORRENT method proposed by Bhatia et al. [2] on both OAA and AAA. TORRENT can resist AAA when the white noise $\epsilon$ is not considered in the model. In order to evaluate the influence of white noise on robust regression, the true data are generated in two ways, one with white noise ($y_i = \mathbf{x}_i^T w^* + \epsilon_i$) and the other without white noise ($y_i = \mathbf{x}_i^T w^*$). Other settings are the same as those in Section 5. The experimental results are shown in Figure 5. Under these two attacks, the performance of TORRENT algorithm is very consistent with that of CRR in both noisy and noiseless settings. TORRENT performs slightly better than CRR in the absence of white noise, as shown in Figure 5(e). However, both CRR and TORRENT perform poorly under AAA. It can be seen that the TRIP and BRHT algorithms are very robust in all cases.

We also consider another leverage point attack (LPA) on data sets. For a point $(\mathbf{x}_i, y_i)$, the leverage value is defined as $h_{ii} = \mathbf{x}_i^T(XX^T)^{-1}\mathbf{x}_i$. In the linear regression, the regression result can be strongly affected by high leverage points[4]. Therefore, if we corrupt those high leverage points, the regression result is more likely to be unstable. If we set the covariant $\mathbf{x}_i$ as iid in $\mathcal{N}(0, I_d)$, then the high leverage points are roughly those points with large norms $\|\mathbf{x}_i\|_2$ since $\frac{1}{n}XX^T$ converges to $I_d$ as $n \to \infty$. According to the above analysis, we set the LPA as follows: choose $k$ points with the largest covariant norm $\|\mathbf{x}_i\|_2$ and set their corresponding $y_i$ to 0. In this experiment, the true coefficient $\mathbf{w}^*$ is chosen to be a random unit norm vector and the covariant $\mathbf{x}_i$ are iid in $\mathcal{N}(0, I_d)$. The true data (before attack) are also generated in two ways, one with white noise $y_i = \mathbf{x}_i^T w^* + \epsilon_i$ and the other without white noise $y_i = \mathbf{x}_i^T w^*$, where $\epsilon_i$ are iid in $\mathcal{N}(0, \sigma^2)$. We set $\sigma = 1$ in the experiments. The experimental results are shown in Figure 6. Under LPA, CRR performs poorly and usually collapses first among these methods. Rob-ULA has relatively better performance when the

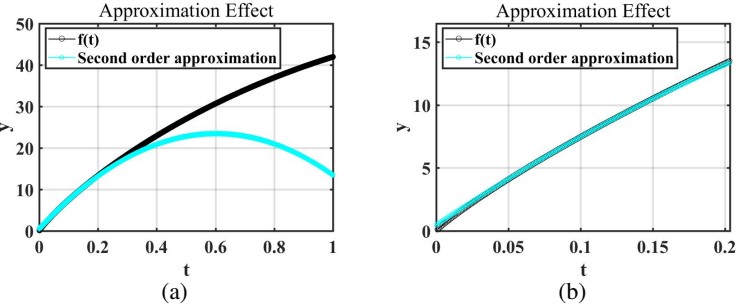

(a)                                          (b)

Figure 3: (a) The approximation of the second order Taylor's expansion on $[0, 1]$. (b) Approximation on the interval $[0, 0.2]$.

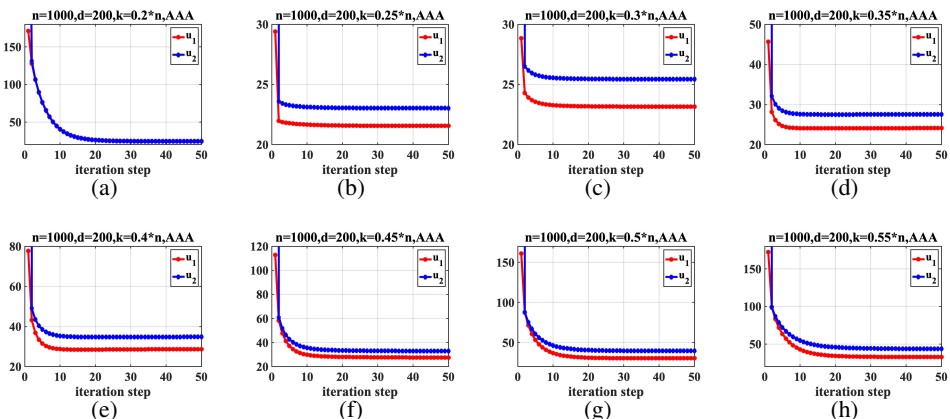

Figure 4: Variation trends of $u_{1t}$ and $u_{2t}$ during the iteration process.

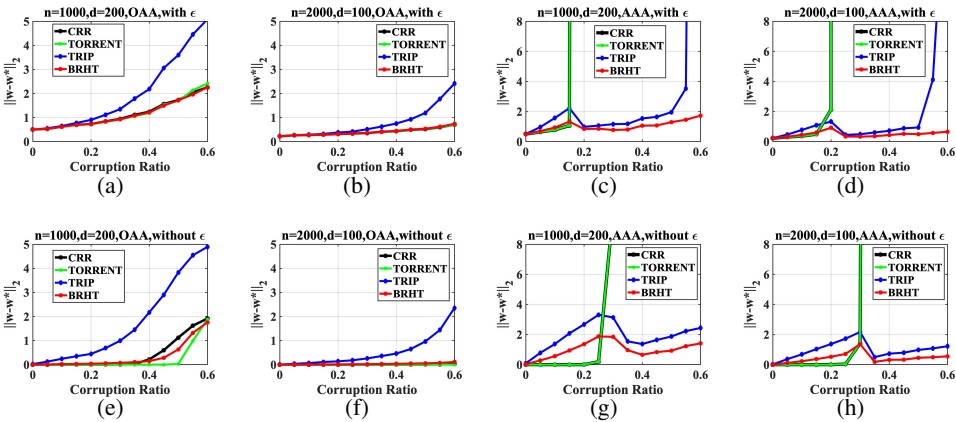

Figure 5: Recovery of parameters with respect to the number of data points $n$, dimensionality $d$, and corruption ratio $\alpha$. (a),(b),(c),(d) consider the case with white noise $\epsilon$, while (e),(f),(g),(h) do not consider white noise. The performance of TORRENT and CRR is similar, and TRIP and BRHT are still more robust under AAAs than CRR and TORRENT.

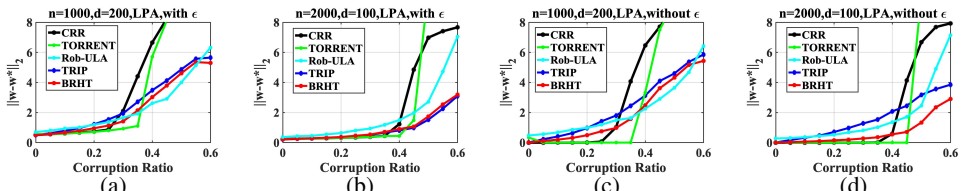

Figure 6: Recovery of parameters with respect to the number of data points $n$, dimensionality $d$, and corruption ratio $\alpha$ under LPA. TRIP and BRHT perform significantly better than CRR. BRHT is more robust in all cases. TORRENT and Rob-ULA show robustness in some cases, but still have limitations.

proportion of outliers is high, but there will be relatively large errors in the case of low proportion of outliers. TORRENT is very robust under LPA, especially in the absence of white noise. However, if the data dimension is high and the sample size is small, TORRENT is easier to collapse. The proposed TRIP and BRHT are still better than CRR, and will maintain a robust result even there are lots of outliers. The estimation errors of BRHT are smaller than TRIP, which shows BRHT is the most robust algorithm in this experiment.