# OpenReview forum: "Robust Bayesian Regression via Hard Thresholding"
_NeurIPS.cc/2022/Conference — NeurIPS 2022 Accept_

### Official Review · Reviewer_eKkz · 2022-07-11

**Rating:** 5
**Confidence:** 3
**Soundness:** 4 excellent
**Presentation:** 3 good
**Contribution:** 3 good

**Summary:**

The authors study the robust least-squares regression (RLSR). The main contribution of this paper is to propose an algorithm that achieves strong results in terms of resisting adaptive adversarial attacks. A theoretical convergence analysis is provided. Extensive experiments have illustrated that their algorithms outperform SOTA methods in terms of both robustness and efficiency.

**Questions:**

It is better to conduct more experiments in more complicated settings.

**Limitations:**

Yes, the authors have adequately addressed the limitations and potential negative societal impact of their work.

**Strengths And Weaknesses:**

Originality: The related works are adequately cited. The novelty of this paper is high. The results on robust least-squares regression in this paper, will certainly help us have a better understating of adversarial attacks and defenses from a theoretical way. I have checked the technique parts and find that the proofs are solid. I think this is a significant contribution to machine learning immunity.

Quality: This paper is technically sound.

Clarity: This paper is clearly written and well organized. I find it easy to follow.

Significance: I think the results in this paper is significant, as explained above.

---

> ### Author Response · Authors · 2022-08-02
> **Response to Reviewer eKkz**
>
> We are thankful for the positive and constructive feedback, especially on checking the technique parts. In the following we briefly respond your question.
>
> **Question1**: It is better to conduct more experiments in more complicated settings.
> **Response**: Thanks for the suggestion, and more experiments have been conducted in more complicated settings. Specifically, in order to better demonstrate the performance of the proposed methods, we have added more experiments to compare with TORRENT algorithm, proposed by Bhatia et al. in 2015 [2]. TORRENT is a robust regression method which is based on a thresholding operator. CRR[1] is another thresholding operator based robust regression algorithm, which will perform better in noisy case compared with TORRENT. OAA and AAA mentioned in Section 5 are still used as attack methods, and the experiment is divided into two cases: with white noise $\boldsymbol{\epsilon}$ and without white noise. Under these two attacks, the performance of TORRENT algorithm is very consistent with that of CRR[1] in both noisy and noiseless settings. TORRENT performs slightly better than CRR in the absence of white noise. However, both CRR and TORRENT perform poorly under AAA. It can be seen that the TRIP and BRHT algorithms are very robust in all cases. The detailed experimental results are shown in **Appendix E**.
> Furthermore, in order to better illustrate the robustness of our methods, the leverage point attack (LPA) is also considered in the experiment as shown in **Appendix E**. Through attacking the high leverage points in the data, LPA can effectively corrupt the regression results. The experimental results show that CRR performs poorly under LPA, and usually collapses first among these methods. Rob-ULA[3] is a bayesian descent method using an unadjusted Langevin algorithm (ULA). Rob-ULA has relatively better performance when the proportion of outliers is high, but there will be relatively large errors in the case of low proportion of outliers. TORRENT is very robust under LPA, especially in the absence of white noise. However, if the data dimension is high and the sample size is small, TORRENT is easier to collapse. The proposed TRIP and BRHT are still better than CRR, and will maintain a robust result even there are lots of outliers. The estimation errors of BRHT are smaller than TRIP, which shows BRHT is the most robust algorithm in this experiment.
>
> [1] Kush Bhatia, Prateek Jain, Parameswaran Kamalaruban, and Purushottam Kar. Consistent robust regression. Advances in Neural Information Processing Systems, 30, 2017.
> [2] Kush Bhatia, Prateek Jain, and Purushottam Kar. Robust regression via hard thresholding. Advances in Neural Information Processing Systems, 28, 2015.
> [3] Kush Bhatia, Yi-An Ma, Anca D Dragan, Peter L Bartlett, and Michael I Jordan. Bayesian robustness: A nonasymptotic viewpoint. arXiv preprint arXiv:1907.11826, 2019.

---

### Official Review · Reviewer_AqFQ · 2022-07-12

**Rating:** 6
**Confidence:** 3
**Soundness:** 3 good
**Presentation:** 3 good
**Contribution:** 3 good

**Summary:**

The paper works on robust least-squares regression of the form $\mathbf{y} = X^{T}\mathbf{w}^* + \mathbf{b}^* + \boldsymbol{\epsilon}$, where $\mathbf{b}^*$ represents a $k$-sparse adversarial perturbation. Specifically, it proposes a Bayesian extension of the Hard Thresholding method previously designed against oblivious adversarial attacks (OAAs), to make it resistant against (a stronger) adaptive adversarial attacks (AAAs), that assumes an adversary having an access to $X$, $\mathbf{w}^*$ and $\boldsymbol{\epsilon}$ before attacks. Two methods are proposed, namely TRIP and BRHT, that incorporate a Gaussian prior on weights $w$ and additionally on exponential weighting $r$, respectively. The paper provides both theoretical convergence analysis and empirical results showing that the proposed methods are more robust under AAAs while also improving under the prior setup of OAAs.

**Questions:**

- Section 3.1: Is the proposed hard-thresholding based method actually induced from assuming Gaussian weight prior, or they are just orthogonal, i.e., TRIP works without the prior? More explanation on the relationship between the prior assumption and TRIP may help.
- For TRIP, I feel it is a bit unclear why one should assume the Gaussian prior for robustness against AAAs, but perhaps not for others? If not, the paper could empirically explore the effect of using different priors on weights.
- Line 115: More discussions could be made on how mild the assumed SSC and SSS conditions in both theoretical and empirical senses?

**Limitations:**

The paper does not include discussions on potential negative societal impact.

**Strengths And Weaknesses:**

**Strengths**

- The paper is clearly-written, e.g., it clearly presents the problem formulation and methodologies.
- The proposed methods are simple, and well-motivated.
- The paper conducts empirical analysis to support the claims and reports improved results. In the experiments, the paper designs an adaptive attack scheme ADCA in a similar manner to TRIP but in the opposite direction in optimization.

**Weaknesses**

- Given that AAA is assumed to be an adversary, a more diverse attack methods could be considered - the current attack method of ADCA is designed to be quite close to TRIP, and it is likely that TRIP performs better than others under ADCA because it is just trained as an opposite of ADCA and may not generalize under other ways of attacks.
- Although it is still novel to me that the paper aims for adversarial robustness, Bayesian approaches on weights and loss coefficients have been quite known approaches in general so some readers may concern on the technical novelty on the method.

---

> ### Author Response · Authors · 2022-08-02
> **Response to Reviewer AqFQ**
>
> We thank the reviewer for the constructive feedback. In the following paragraphs we briefly respond the questions, and details can be found in the rebuttal revision.
>
> **Weakness 1**:	Given that AAA is assumed to be an adversary, a more diverse attack methods could be considered - the current attack method of ADCA is designed to be quite close to TRIP, and it is likely that TRIP performs better than others under ADCA because it is just trained as an opposite of ADCA and may not generalize under other ways of attacks.
> **Response**: Thank you for the comment. Previous work rarely consider the attacks that all the information in the model is known. Most of the attacks considered are OAA or attacks that only rely on part of the model information. Therefore we propose the ADCA algorithm in order to verify that our proposed method TRIP and BRHT can also have a certain robustness in the most complex cases. In order to better illustrate the robustness of our methods, the leverage point attack (LPA) is also considered in **Appendix E**. Through attacking the high leverage points in the data, LPA can effectively corrupt the regression results. The detailed experimental results are shown in **Appendix E**, which reflect that TRIP and BRHT are also robust compared with other methods.
>
>
> **Weakness 2**:	Although it is still novel to me that the paper aims for adversarial robustness, Bayesian approaches on weights and loss coefficients have been quite known approaches in general so some readers may concern on the technical novelty on the method.
> **Response**: Thanks for comment. We think the novelty lies in the combination of Bayesian method and the hard thresholding operator with better performance. The use of hard thresholding operator in robust regression was first proposed by Bhatia[2] in 2015, and our paper may be the first one that combines Bayesian method with the hard thresholding operator in the published literature. The main purpose of our paper is to show the prior information can make regression more robust, and Bayesian method is an effective tool to integrate prior information. This article is an attempt in this direction, and we demonstrate that this combination can greatly improve the effect of robust regression. This method could be potentially useful in practical application scenarios.
>
> **Questions 1&2**:(1) Section 3.1: Is the proposed hard-thresholding based method actually induced from assuming Gaussian weight prior, or they are just orthogonal, i.e., TRIP works without the prior? More explanation on the relationship between the prior assumption and TRIP may help. (2) For TRIP, I feel it is a bit unclear why one should assume the Gaussian prior for robustness against AAAs, but perhaps not for others? If not, the paper could empirically explore the effect of using different priors on weights.
> **Response**:Thanks for the suggestions. If we do not add prior on TRIP, this method is actually CRR[1]. The Gaussian prior itself is not proposed from the aspect of robustness, but because Gaussian prior and the regression likelihood are conjugate. This conjugation makes it possible to reduce the amount of computation in the iteration of TRIP and BRHT algorithms. For TRIP, an explicit iterative form can be derived to make its calculation more convenient, and the theoretical proof can also be presented in a relatively simple form. And for BRHT, the calculation steps can also be reduced during the iteration of VBEM algorithm. Other priors may achieve similar results, but they will cause difficulties in calculation, and reduce the efficiency of the algorithm. In future research, we will also consider whether we can introduce a more effective prior, which can enhance robustness and reduce the calculation.
>
> **Question 3**:	Line 115: More discussions could be made on how mild the assumed SSC and SSS conditions in both theoretical and empirical senses?
> **Response**: Thanks for the constructive suggestion. We add a theoretical description of these two conditions in **Appendix C.1**. These two conditions are common in robust regression methods based on hard thresholding operators, such as [1],[2]. Because these two conditions are mainly used in theoretical proof, we pay more attention to the theoretical bound of these two conditions. Under the properties of these two conditions, we can prove that TRIP has a non-zero breakdown point when  $\mathbf{x}_i$ are iid in $\mathcal{N} (0,\Sigma)$, which also shows that these two conditions are not so strict.
>
>
> [1] Kush Bhatia, Prateek Jain, Parameswaran Kamalaruban, and Purushottam Kar. Consistent robust regression. Advances in Neural Information Processing Systems, 30, 2017
> [2] Kush Bhatia, Prateek Jain, and Purushottam Kar. Robust regression via hard thresholding. Advances in Neural Information Processing Systems, 28, 2015.

---

> > ### Comment · Reviewer_AqFQ · 2022-08-09
> > **Post-rebuttal update**
> >
> > Thanks for the response. I appreciate the authors provide additional experiments as well as more clarification in the revision. I would keep my original evaluation leaning toward acceptance.

---

### Official Review · Reviewer_DbYe · 2022-07-12

**Rating:** 7
**Confidence:** 3
**Soundness:** 3 good
**Presentation:** 3 good
**Contribution:** 2 fair

**Summary:**

The paper studies the problem of robust linear regression and demonstrates two algorithms which improves the breakdown point for this algorithm under adversarial attacks compared to prior work.

The main contribution is the observation that, with some prior information, even if it is far from the truth, it is possible to improve the breakdown point and find a consistent estimator for the problem. Prior work was able to find consistent estimators only in the setting where the corruption vector was chosen oblivious to the measurements.

The authors consider the Adaptive Adversarial Attacks (AAA)  setting where the observations are given by $(X, y)$, where $X \in \mathbb{R}^{d \times n}$ and $y \in \mathbb{R}^n$. The observations satisfy $y = Xw^* + b^* + \epsilon$ where $w^*$ is the true regression coefficient, $\epsilon$ is the dense noise and $b^*$ is an arbitrary $k^*$-sparse vector. The goal is to recover $(\hat{w}, \hat{S})$ where $\hat{w}$ is close to $w^*$ and $\hat{S}$ is an estimate of the uncorrupted sample set.  The authors assume that the variance of the noise $\epsilon$ can be controlled by the algorithm designer and be set to $\sigma$, or can be estimated independently to a high degree of accuracy.

The first algorithm (TRIP) assumes the prior on $w*$ is $\mathcal N(w_0, \Sigma_0)$ for some $w_0$ and $\Sigma_0$ determined in advance. This essentially leads to the problem being transformed into a regularised least squares problem, which they solve by a hard-thresholding approach similar to [1]. The second algorithm is more complicated and assumes a prior also on the local weight assigned to specific samples and uses variational Bayesian expectation maximization to solve a reweighted probabilistic model for linear regression.


**Questions:**

1. It would be good to explicitly write down the breakdown point achieved in the case of a Gaussian. For instance [2] explicitly says that one lower bound for the breakdown point is 1/65.
2. I would be curious to see how this algorithm performs with the following noise model from [3]: Samples $(x, y)$ where $x \sim N(0, 1)$ and $y = 100 x$, where the adversary looks at the $\epsilon$ fraction of the $x_i$ such that $|x_i|$ is maximized, and then sets the corresponding $y$ to 0. The plot of the error $\| \widehat w - w^* \|$ vs $\epsilon$ might shed light on the comparison between breakdown points in the case of a Gaussian. While I understand that the goal is to provide a consistent estimator, It might be clearer to see the performance of the algorithm and the breakdown point for an example with no additive error.
3. Could the authors please explain for my understanding, intuitively, why the existence of the prior allows one to solve the problem in the harder adversarial setting? Somehow this has still escaped me.

I will reconsider my score depending on the answers to these questions.

-----

Update: The authors have addressed all questions raised, and I have updated the score to reflect my new assessment. I still have some concern about the theoretical breakdown point, but the algorithms seem to perform well on noise models which previous algorithms fail on.

**Limitations:**

This is a theoretical result with limited societal impact.

**Strengths And Weaknesses:**

Strengths:

It is qualitatively interesting that bayesian updates can lead to consistent estimators in the AAA setting. I like the result.


Weaknesses:

1. I feel that the authors need to place the result in the context of the surrounding literature a little better. This paper considers the model where only the labels are corrupted (i.e. only $y$) while the result by Diakonikolas et al. [6] (mentioned on line 53) is for the setting where both the $X$ as well as $y$ values might be corrupted.

2. Additionally, further comparison with [2] would be helpful, since the goal of this paper seems to be to achieve a good breakdown point for the AAA setting, which I feel [2] also addresses, as opposed to [1] which only deals with the oblivious setting (with the additional goal of being a consistent estimator, i.e. if the dense additive noise is mean 0, the goal here is for the estimator to achieve error going to 0 as the number of samples tends to infinity).

3. Another paper to cite might be https://arxiv.org/abs/1809.08055, which attempts to solve the problem using L1-regression but it appears this is not a consistent estimator.

4. I found the paper a little hard to understand. Clearer motivation and a description of the algorithm would have been helpful. I would also have liked there to be more intuition and a comparison of the difference in terms of the algorithms and updates from [1] and [2] that would help the reader better understand how the assumption of the prior helps.

[1] Kush Bhatia, Prateek Jain, Parameswaran Kamalaruban, and Purushottam Kar. Consistent robust regression. Advances in Neural Information Processing Systems, 30, 2017.

[2] Kush Bhatia, Prateek Jain, and Purushottam Kar. Robust regression via hard thresholding. Advances in Neural Information Processing Systems, 28, 2015.

---

> ### Author Response · Authors · 2022-08-02
> **Response to Reviewer DbYe**
>
> We thank the reviewer for the constructive feedback. The suggestions and questions are briefly responded below, and details can be found in the rebuttal revision.
>
> **Weakness 1**: I feel that the authors need to place the result in the context of the surrounding literature a little better...
> **Response**: Thanks for your suggestion. We have added the description of relevant literature and adjusted the expression of the other literature. The changes are made in **lines 53 to 59** of the rebuttal revision.
>
>
> **Weakness 2** : Additionally, further comparison with [2] would be helpful...
> **Response**: We add an experiment to compare with TORRENT[2] under OAA and AAA. The experiment is divided into two cases: with white noise $\boldsymbol{\epsilon}$ and without white noise. Under attacks of OAA and AAA, the performance of TORRENT is similar with CRR[1]. Operation details and experimental results are shown in **Appendix E**.
>
>
> **Weakness 3**:Another paper to cite might be https://arxiv.org/abs/1809.08055...
> **Response**: This is a closely related paper and has been cited as **[10]**. Thanks for the suggestion.
>
> **Weakness 4**: Clearer motivation and a description of the algorithm would have been helpful...
> **Response**: We have added more explanations on TRIP and explained the differences between it and the previous methods. This part is described from **line 154 to 161**. Here we mainly compare our methods with CRR. Since the idea of TRIP is very different from TORRENT, they are not compared.
>
> **Question 1**: It would be good to explicitly write down the breakdown point achieved in the case of a Gaussian...
> **Response**: Thanks for your inspiring advice. We re-order the theorems and add Theorem 3 in **Appendix C.2, line 490**. Theorem 3 provides the conditions that the breakdown point of TRIP algorithm should satisfy, but this theorem is not very intuitive and cannot give the explicit form of the breakdown point. Therefore, following this theorem, we give an approximate expression of the breakdown point, which is based on a second order Taylor expansion.
> >Suppose $\lim_{n\to \infty}\frac{\lambda_{min}(M)}{n}=\xi$, then when $\xi$ is not too large, the approximate expression of the breakdown point is $k^{*}\le k\le (0.3023-\sqrt{0.0887-0.0040\xi})n$. It can be seen that as the weight of prior gradually increases, the breakdown point gradually rises as well.
>
> **Question 2**: I would be curious to see how this algorithm performs with the following noise model from [3]...
> **Response**: Thanks for your comment. The mentioned attack can actually be regarded as a leverage point attack, that is, attacking the high leverage points in all samples. This attack can effectively corrupt the regression results. We extend this attack to high dimensions to verify the robustness of the methods. This experiment is also divided into two cases: with and without white noise. Under the attack mentioned above, TRIP and BRHT proposed in this paper are compared with CRR, TORRENT, and Rob-ULA. The analysis and experimental results of this attack are shown in **Appendix E**. The experimental results show that TORRENT indeed achieves good regression results when white noise is not considered. However, when the data dimension is high and the samples size is relatively small, TRIP and BRHT are more robust.
>
> **Question 3**: Could the authors please explain for my understanding, intuitively, why the existence of the prior allows one to solve the problem in the harder adversarial setting...
> **Response**:  The difference between the proposed TRIP and the original CRR [1] is the form of iteration step. The iteration step in both TRIP and CRR can be expressed uniformly as $HT_k(\mathbf{y}-X^T\mathbf{w}^t)$, but $\mathbf{w}^t=(XX^T+M)^{-1}[X(\mathbf{y}-\mathbf{b}^t)+M\mathbf{w}_0]$ in TRIP and $\mathbf{w}^t=(XX^T)^{-1}X(\mathbf{y}-\mathbf{b}^t)$ in CRR. The $\mathbf{w}^t$ in CRR is just a simple least square estimation, while adding the prior in TRIP is equivalent to adding a quadratic regularization in each iteration. This quadratic regularization can avoid the candidate of iteration that is too far from the prior mean, which is also helpful to ensure the numerical stability of solution. Thus, as long as the prior is not mis-specified too much, TRIP will be more likely to identify the uncorrupted points, and the final result of TRIP will be more robust than CRR. Compared with TRIP, BRHT improves robustness in each iteration. As a result even if the prior weight is low, each iteration of BRHT can try to avoid the influence of outliers and find points that have not been corrupted within a larger search scope.
>
> [1] Kush Bhatia, Prateek Jain, Parameswaran Kamalaruban, and Purushottam Kar. Consistent robust regression. Advances in Neural Information Processing Systems, 30, 2017.
> [2] Kush Bhatia, Prateek Jain, and Purushottam Kar. Robust regression via hard thresholding. Advances in Neural Information Processing Systems, 28, 2015.

---

> > ### Comment · Reviewer_DbYe · 2022-08-03
> > **Update**
> >
> > Thank you for addressing my questions.
> >
> > It appears that the theoretical breakdown point is much smaller than 1/65 $\approx$ 0.0153, or 0.23 for reasonable values of $\xi$, is that right? Given that the objective of the paper is to demonstrate a high breakdown point, this is a little unfortunate. Am I reading that wrong? Can the breakdown point be as large as 0.3023?
> >
> > But it does look like in all the experiments the breakdown point is much higher than previous algorithms, so it is possible that this is an artefact of the proof.
> >
> > It would have been nice to have tight results here (i.e. a noise model where a high breakdown point is achieved, and a theoretical upper bound that matches it).

---

> > > ### Author Response · Authors · 2022-08-04
> > > **Response to Reviewer DbYe**
> > >
> > > Thank you for the further comment and for raising our score!   We want to address our thoughts on the breakdown point as follows.
> > >
> > > Firstly, the explicit form of the breakdown point we have shown in the rebuttal is an approximate result when $\xi$ is not large. Theorem 3 accurately describes the breakdown point of TRIP. If we do not add prior to TRIP, this method is actually CRR[1],
> > >  and the breakdown point is the lowest in this case. From the perspective of Theorem 3, the breakdown point will increase monotonically as $\xi$ increases and it could be as large as 0.3023. Actually, if $\xi$ reaches 49, the breakdown point of TRIP will be 1 theoretically (although the solution at this time is basically determined by the prior).
> > >
> > > Secondly, the theoretical breakdown point is much smaller than that in the experiments and practical applications, because the proof needs lots of inequality scaling. TORRENT[2] has a breakdown point of 1/65 in the noiseless case, and has no theoretical guarantees in the noisy case. The theoretical breakdown point of CRR is 1/10000 in noisy case (although their proof of error bound is more rigorous than ours), but the practical effect is excellent. Under the same conditions, the breakdown point of our methods is better than CRR. Although from the perspective of Theorem 3, it seems that we need to add a relatively high weight on prior to reach a high breakdown point. However, this phenomenon is also due to the scaling of inequalities, and practical applications may only require a low weight on prior to achieve good results.

---

### Meta-Review · Area_Chair_McBT · 2022-08-29

**Recommendation:** Accept
**Confidence:** Certain

**Metareview:**

The paper studies the problem of label-outlier robust regression with prior on the optimal parameter. The reviewers agree that the results are novel and significant. There is certainly a concern about the novelty of the method and about additional insights provided by the result. However, as the paper studies this relatively new problem and provides solid results for it, we  recommend accepting it for publication.

**Award:**

No

---

### Decision · Program_Chairs · 2022-09-14

Accept